# Interpreting and Enhancing Emotional Circuits in Large Vision-Language Models via Cross-Modal Information Flow

**Chengsheng Zhang** [1]   **Chenghao Sun** [1 2]   **Zhining Xie** [2]   **Xinmei Tian** [1]

## Abstract

Large Vision-Language Models (LVLMs) represent a significant leap towards empathetic agents, demonstrating remarkable capabilities in emotion understanding. However, the internal mechanisms governing how LVLMs translate abstract visual stimuli into coherent emotional narratives remain largely unexplored, primarily due to the scarcity of visual counterfactuals and the diffuse nature of emotional expression. In this paper, we bridge this gap by introducing a steering-vector-based causal attribution framework tailored for descriptive emotional reasoning. To this end, we construct a specialized dataset to demystify the emotional circuits underlying the three-stage "Adapt-Aggregate-Execute" mechanism. Crucially, we discover a functional decoupling: visual emotional cues are aggregated in middle layers via *sentiment-specific* attention heads, but are subsequently translated into narrative generation in deep layers through *emotion-general* pathways. Guided by these insights, we regulate the emotional information routing to strengthen attention flow and amplify the semantic activation to consolidate expression. Extensive experiments on the comprehensive MER-UniBench demonstrate that our methods significantly improve performance via inference-time intervention, effectively mitigating emotional hallucinations and corroborating the causal fidelity of the discovered circuits.

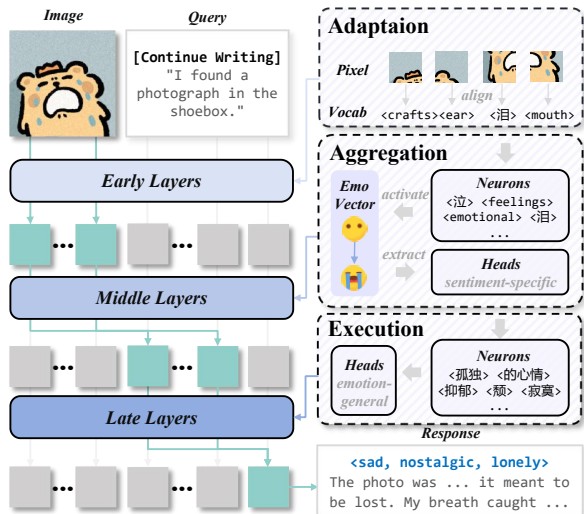

*Figure 1.* The overview of emotional mechanisms in LVLM, which involves: **(1)** adapting the image modality, **(2)** aggregating emotional intention, and **(3)** executing emotional expression.

## 1. Introduction

Large Vision-Language Models (LVLMs) are evolving from static perceivers to empathetic agents, promising transformative impacts in human-computer interaction (Liu et al., 2021) and embodied intelligence (Spezialetti et al., 2020). However, current LVLMs remain prone to *emotional hallucinations* (Xing et al., 2025; Zhao et al., 2025; 2023)—generating linguistically coherent narratives that contradict the affective content of visual stimuli (*e.g.*, describing a weeping face as expressing joy). Unlike object hallucinations (Li et al., 2023; Rohrbach et al., 2018) which lead to factual errors, such affective misalignment risks violating social norms and ethical boundaries (Jiao et al., 2025; Sharma et al., 2023). Despite recent advancements in visual instruction tuning (Xie et al., 2024; Cheng et al., 2024) and reinforcement learning (Zhao et al., 2025; Lian et al., 2025b), these black-box and data-driven optimization techniques merely fit behaviorist fine-tuning without guaranteeing internal alignment (Ngo et al., 2022; Lu et al., 2025). Consequently, to fundamentally address these reliability bottlenecks, it is imperative to scrutinize the *internal mechanisms* governing how LVLMs perceive and express emotion.

---

[1]MoE Key Laboratory of Brain-inspired Intelligent Perception and Cognition, University of Science and Technology of China [2]AIPD, Tencent. Correspondence to: Chenghao Sun <chsun@mail.ustc.edu.cn>, Xinmei Tian <xinmei@ustc.edu.cn>.

*Proceedings of the $43^{rd}$ International Conference on Machine Learning*, Seoul, South Korea. PMLR 306, 2026. Copyright 2026 by the author(s).

While mechanistic interpretability in LLMs has successfully localized critical components responsible for emotion processing (Tak et al., 2025; Lee et al., 2025), the internal emotional circuitry of LVLMs remains unexplored. Existing LVLM interpretability research predominantly targets *object perception*, focusing on the causes of object hallucinations (Jiang et al., 2025) and cross-modal alignment (Neo et al., 2025). In contrast, it remains unclear how LVLMs translate specific objects into affective semantics and utilize them to guide the emotional expression.

Bridging this gap presents significant challenges, as current methodologies are ill-equipped for emotion-centric causal analysis. First, the absence of counterfactual visual benchmarks. Unlike in LLMs, where implemented via lexical substitution (*e.g.*, replacing "happy" with "sad") (Tak et al., 2025), it remains a formidable obstacle to isolate emotional attributes without altering narrative content. Second, the ineffectiveness of discrete metrics. Emotional expression is inherently *diffusive* (Frijda et al., 1989), manifesting through the holistic tone of long-form generation rather than a single word. Thus, standard *Next-Token-Prediction* metrics (Zhang et al., 2025) fail to capture these affective subtle shifts.

To dismantle these barriers, we propose a *Steering-Vector-based Causal Attribution Framework* tailored for descriptive reasoning. First, we construct semantically controlled visual counterfactuals. Through strictly paired emotional and neutral contrasts, we decouple the emotional semantics from visual features, enabling the extraction of various emotional directions (see Figure 2, Stage-I). Second, leveraging these directions as probes, we design a *Latent Restoration Metric* to quantify the causal effect of distinct modules, shifting the evaluation anchor from discrete token probabilities to hidden spaces. Guided by this metric, we implement a hierarchical discovery strategy. Starting from pivotal layers, we recursively trace the critical attention heads aggregating emotional information and the MLP neurons encoding emotional features, thereby demystifying the cross-modal emotion flow (see Figure 2, Stage-II).

Applying this framework, we uncover a clear "Adapte-Aggregate-Execute" emotion mechanism (see Figure 1) in LVLMs. In the Shallow Layers (*Adaptation*), visual features undergo modality alignment to bridge the semantic gap. Subsequently, in the Middle Layers (*Aggregation*), the model activates *Contextual Trigger Neurons* (encoding situational cues) and aggregates these signals into the Query token (acting as a visual summarizer) via *emotion-specific heads*—which are selectively active for distinct emotion categories. Finally, transitioning to the Deep Layers (*Execution*), the Query token activates *Explicit State Neurons* (encoding the emotion itself) and steers the narrative generation via *emotion-general heads* that function universally across emotions. This functional decoupling between in-

formation routing and semantic activation lays the critical groundwork for our targeted intervention.

Inspired by these insights, we propose VEENA (see Figure 3), a training-free, surgical inference-time intervention framework. Synergistically aligning with the emotion mechanism, VEENA comprises: *Visual Emotion Enhancement* (*VEE*) regulates the information routing to strengthen attention flow, while *Emotional Neuron Augmentation* (*ENA*) amplifies the semantic activation to consolidate expression. Extensive experiments on MER-UniBench (Lian et al., 2025a) validate its effectiveness: VEENA significantly improves performance without additional latency. This substantial improvement not only effectively mitigates emotional hallucinations but also strongly corroborates the causal fidelity of our discovered emotional circuits.

Our main contributions are summarized as follows:

- We propose a vector-based attribution framework, transcending discrete metric limitations to enable the first precise localization of emotional circuits in LVLMs.

- We demystify the "Adapte-Aggregate-Execute" emotion mechanism, delineating the cross-modal flow from shallow modality adaptation and middle-layer intent aggregation to deep-layer concept execution.

- We introduce VEENA, a surgical, training-free intervention. Extensively experiments demonstrate its efficacy, corroborating the fidelity of our mechanistic findings.

## 2. Preliminaries

**Descriptive MER Evaluation.** Typically, the Multi-modal Emotion Recognition (MER) framework comprises a visual input (*e.g.*, an image) paired with a corresponding query, which the model is tasked with predicting the emotion category from limited label spaces (*e.g.*, *happy*). Specifically, let $X = \text{Concat}(I, T)$ denote an input sequence with its ground-truth emotion label $y$, where $I$ and $T$ represent visual and textual token embeddings, respectively. The model performs *Next-Token-Prediction* (*NTP*) to maximize the likelihood $P(y|X)$, evaluated by classification accuracy. However, considering the rich spectrum of human emotions, recent research introduces (Lian et al., 2025a;b) open-ended metrics to transcend the limitations of fixed label spaces. Adopting this protocol, given the response sequence $R_i$ generated by the model, we employ an *LLM-as-Judge* evaluator to extract a set of $N_p$ emotional keywords $\mathbf{Y} = \{\hat{y}_j\}_{j=1}^{N_p}$ that characterize the sentimental tone of the narrative. Subsequently, we use a hierarchical mapping function $\Phi(\cdot)$, which projects free-form keywords onto $N_w$ standardized emotion wheels (Lian et al., 2025a), and calculate the *hit*

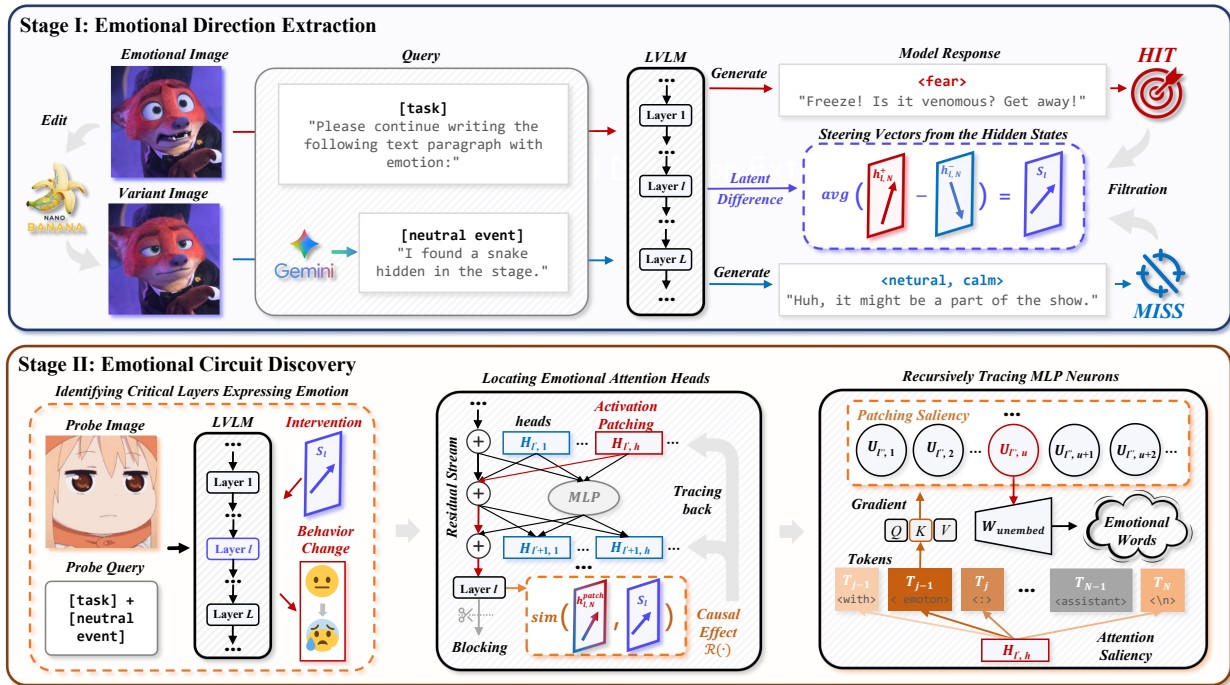

*Figure 2.* Overview of our mechanistic interpretability framework. **Stage I:** We construct contrastive input pairs (emotional vs. neutral) to extract generalized emotional steering vectors from hidden states, filtering based on hit rate. **Stage II:** Adopting a coarse-to-fine localization strategy, we first identify critical emotion layers, then pinpoint specific attention heads, and recursively trace MLP neurons.

*rate* metric $\mathcal{H}(\cdot, \cdot)$ as follows:

$$\mathcal{H}(X, y) = \frac{1}{N_w} \sum_{k=1}^{N_w} \mathbb{I}\left[\Phi(y) \in \Phi(\mathbf{Y})\right] \quad (1)$$

where $\mathbb{I}[\cdot]$ is an indicator function.

**Causal Analysis via Activation Patching.** To identify the critical components of internal mechanisms, we leverage a mechanistic interpretability technique, *Activation Patching* (Golovanevsky et al., 2024; Li et al., 2025a), which builds upon *NTP*. Consider a contrastive pair: a *positive* input $X^+$ that elicits the correct answer $y^+$, and a *negative* input $X^-$ with a different answer $y^-$. We measure the performance of the model by calculating the *Logit Difference* $\mathcal{F}(X)$ between the target answer $y^+$ and the counterfactual answer $y^-$ to quantify the model's reasoning confidence:

$$\mathcal{F}(X) = \text{Logit}(y^+|X) - \text{Logit}(y^-|X). \quad (2)$$

To measure the causal effect of component $c$, we define the patched utility $\mathcal{F}_{\text{patch}}$ by intervening on the model's computation graph. Specifically, we execute the model on $X^-$ while patching the activation of $c$ to assume activation value $A_c^+$ recorded from the positive input $X^+$. The normalized causal effect $\mathcal{E}(c)$ is then derived as:

$$\mathcal{E}(c) = \frac{\mathcal{F}_{\text{patch}}(X^- \leftarrow A_c^+) - \mathcal{F}(X^-)}{\mathcal{F}(X^+) - \mathcal{F}(X^-)}, \quad (3)$$

where $\mathcal{F}_{\text{patch}}(X^- \leftarrow A_c^+)$ represents the logit difference obtained under the intervention. A value of $\mathcal{E}(c) \approx 1$ indicates

that component $c$ is sufficient to restore the correct prediction from the counterfactual state. Nonetheless, this token-centric metric is ill-suited for the descriptive paradigm, as the open-ended nature of the generated narratives precludes the definition of a deterministic target token required for computing the rigid logit difference $\mathcal{F}(X)$.

## 3. Methodology

### 3.1. Emotional Direction Extraction

We utilize activation steering (Liu et al., 2024a; Li et al., 2025b) to isolate emotional directions from the model's hidden states (Figure 2 Stage-I). To this end, we construct a contrastive dataset $\mathcal{D}$ using paired context sequences (see Section 4). Given an image $I_{emo}$ with emotion label $y_i$ (*e.g.*, *fear*), we define a *positive* input sequence $X^+ = \text{Concat}(I_{emo}, T_{neu})$ and its counterpart $X^- = \text{Concat}(I_{neu}, T_{neu})$, where $T_{neu}$ represents the query combined the task prompt with a randomly selected neutral event. Subsequently, we perform generation passes on both sequences and extract the residual stream activations at the last input token positions $N$. The emotional direction vectors $s_{i,l}$ for the $i$-th pairs at layer $l$ is computed as the difference between the activations $h_{i,l,N}$ as:

$$s_{i,l} = h_{i,l,N}^+ - h_{i,l,N}^-. \quad (4)$$

To eliminate image-specific semantics and capture a generalized emotional direction, we compute the global steering

vector $S_l$ by averaging the directions over a filtered subset of pairs. Crucially, we only include pairs where the model successfully exhibited the target emotion:

$$S_l = \frac{1}{|\mathcal{U}|} \sum_{i \in \mathcal{U}} s_{i,l}, \quad \mathcal{U} = \{i \in \mathcal{D} \mid \mathcal{H}(X_i^+, y_i) > \tau\}. \quad (5)$$

Here, $\mathcal{U}$ denotes the valid steering set, and $\tau$ is the threshold.

### 3.2. Emotional Circuit Discovery

As shown in Figure 2 Stage-II, we employ a hierarchical discovery strategy: 1) identifying critical layers via causal intervention, 2) locating heads via activation patching, and 3) tracing the information flow back to upstream neurons.

**Identifying Critical Layers Expressing Emotion.** We first assess the causal effect of each layer by injecting the extracted emotional directions during inference. Given a *probe negative* input sequence $X_j^-$, we modify the hidden states of layer $l$ as:

$$\tilde{h}_{j,l,t}^- = h_{j,l,t}^- + \alpha \cdot S_l, \quad (6)$$

where $\alpha$ is a scalar coefficient controlling intervention strength. We quantify the induced behavioral impact by comparing the post-intervention hit rate $\tilde{\mathcal{H}}$ from complete responses with the baseline $\mathcal{H}$. The relative change ratio is defined as $\mathcal{C} = (\tilde{\mathcal{H}} - \mathcal{H})/\mathcal{H} \times 100\%$. Layers exhibiting significant $\mathcal{C}$ scores are deemed critical.

**Locating Emotional Attention Heads.** To identify the upstream attention heads (at layer $l' < l$) whose outputs significantly align with the global emotional direction $S_l$, we employ a backward activation patching strategy. Firstly, we calculate the *Emotional Intention* of the model as:

$$\mathcal{I}(A_c) = \text{sim}(A_c, S_l) = \frac{A_c \cdot S_l}{\|A_c\| \|S_l\|}, \quad (7)$$

where $\text{sim}(\cdot, \cdot)$ denotes the cosine similarity function. This shift from discrete token probabilities to hidden spaces is tailored for evaluating descriptive tasks. Then, we execute the model on the $X^-$ but patch the activation of each head with its value recorded from $X^+$. Formally, we define the *Latent Restoration Metric* $\mathcal{R}(H_{l',h})$ as:

$$\mathcal{R}(H_{l',h}) = \frac{\mathcal{I}_{\text{patch}}(O_{attn,l}^- \leftarrow A_{H_{l',h}}^+) - \mathcal{I}(O_{attn,l}^-)}{\mathcal{I}(O_{attn,l}^+) - \mathcal{I}(O_{attn,l}^-)}, \quad (8)$$

where $A_{H_{l',h}}$ is the activation of the $h$-th head $H_{l',h}$ at an upstream layer $l'$, and $O_{attn,l}$ denotes the output of the attention module at the critical layer $l$. A high $\mathcal{R}(\cdot)$ score indicates that a head is a primary driver in steering the representation towards the target emotion.

**Recursively Tracing MLP Neurons.** To pinpoint the granular origin of emotional features, we trace the information

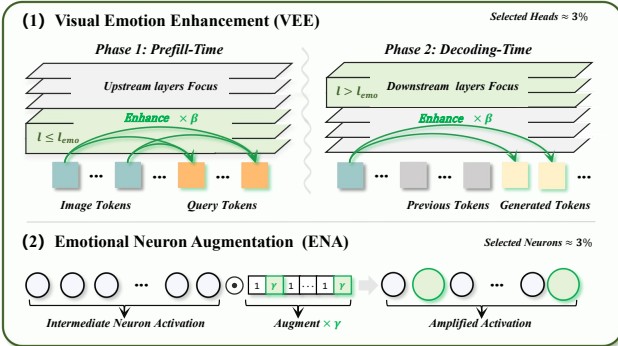

*Figure 3.* The pipeline of our VEENA, which comprises: **(1) VEE** reinforces the flow of visual information to distinct positions to strengthen emotion propagation, and **(2) ENA** amplifies the activation levels of emotion-salient neurons.

flow from upstream MLP neurons to the critical attention heads identified above. For a critical head $H_{l',h}$, we first identify the source token index $t^*$ with the highest attention attribution. Then, we employ *Attribution Patching*, a first-order Taylor-series approximation of activation patching, to approximate the causal effect of upstream neurons without brute-force sweeping. Specifically, we backpropagate the gradients from the *Key projection* $\mathbf{W}_K$ of the critical head to the output of neurons in upstream layers $l'' < l'$. The attribution score $\mathcal{G}$ for a neuron $u$ is formulated as the dot product of its activation difference and the gradient of the similarity metric:

$$\mathcal{G}(U_{l'',u}) = (A_{U_{l'',u}}^+ - A_{U_{l'',u}}^-) \cdot \frac{\partial \mathcal{R}}{\partial A_{U_{l'',u}}^-}$$
$$\approx \Delta A_{U_{l'',u}} \cdot \left( \mathbf{W}_{down,u}^\top \cdot \mathbf{W}_{K,h}^\top \cdot \frac{\partial \mathcal{R}}{\partial \mathbf{k}_{l',h,t^*}} \right), \quad (9)$$

where $U_{l'',u}$ denotes $u$-th neuron at the upstream layer $l''$, $\mathbf{W}_{down,u}$ is the corresponding column in the MLP down-projection matrix, and $\mathbf{k}_{l',h,t^*}$ represents the Key vector of the target head at the source position.

### 3.3. VEENA: Boosting Emotional Intelligence in LVLM

Based on the emotional circuits components and mechanism, we propose *VEENA*, a precise, train-free, inference-time intervention framework, shown in Figure 3.

**Visual Emotion Enhancement (VEE).** We first construct a unified critical head set $\mathcal{C}_{head}$ by aggregating the top-$K_{head}$ heads across emotion categories. Acknowledging the functional distinction between upstream feature extraction and downstream semantic integration, we design a *flow-aware* attention scaling strategy based on the identified critical middle layer $l_{emo}$ (see Section 4.2). Specifically, we define the conditions $\mathcal{P}$, which denoted as:

$$\mathcal{P} = \begin{cases} \mathcal{P}_{up} \triangleq (t=0) \& (l \le l_{emo}) \& (V \to Q) \\ \mathcal{P}_{down} \triangleq (t>0) \& (l > l_{emo}) \& (V \to L) \end{cases} \quad (10)$$

where $t$ is the time step, $Q$ and $L$ are the query and last token, respectively. Crucially, $\mathcal{P}\_up$ triggers during the prefill phase ($t = 0$) at upstream layers, amplifying $V \to Q$ attention to facilitate the Aggregation of visual emotional cues into the textual query. Conversely, $\mathcal{P}\_down$ activates during the decoding phase ($t > 0$) at downstream layers, strengthening $V \to L$ attention to ensure the narrative Execution remains firmly grounded in fine-grained visual details.

Let $W_{l,h} \in \mathbb{R}^{N \times N}$ denote the pre-softmax attention scores for head $(l, h) \in \mathcal{C}_{head}$, we apply an element-wise scaling mask $\mathbf{M}_{head}$ as:

$$\mathbf{M}_{head} = \begin{cases} \beta & \text{if } \mathcal{P}_{up} \vee \mathcal{P}_{down} \\ 1 & \text{otherwise} \end{cases} \tag{11}$$

where $\beta > 1$ is the enhancement coefficient. We correct the original attention weights as $\tilde{W}_{l,h} = W_{l,h} \odot \mathbf{M}_{head}$. Our VEE ensures that during the prefill phase, visual emotion cues are strongly routed to textual queries in upstream layers, while during decoding, emotion consistency is enforced in downstream layers for every generated token.

**Emotional Neuron Augmentation (ENA).** While VEE modulates information routing, ENA explicitly amplifies the emotion-related semantic knowledge stored within the MLP blocks. We aggregate the top-$K_{neuron}$ critical neurons located by each head into a set $\mathcal{C}_{neuron}$. Let $O_{u,l}$ denote the intermediate neuronal activations of the MLP at layer $l$. We augment the expression of emotional features by applying a sparse scaling mask $\mathbf{M}_{neuron}$ as:

$$\mathbf{M}_{neuron} = \begin{cases} \gamma & \text{if } (l, u) \in \mathcal{C}_{neuron} \\ 1 & \text{otherwise.} \end{cases} \tag{12}$$

where $\gamma > 1$ is the excitation coefficient. The modified activations $\tilde{U}_{u,l} = U_l \odot \mathbf{M}_{neuron}$ are then projected by the output weight matrix to form the final MLP output.

## 4. Experiments

### 4.1. Experimental Setup

**Dataset.** We categorize our experimental data into three distinct groups: **(1) Mechanistic Analysis Dataset.** We construct a specialized dataset to facilitate mechanistic discovery and isolate emotion-specific circuits. Specifically, we curate images exhibiting distinct emotional expressions and employ Google Nano Banana to neutralize these images. We prioritized semantic content preservation during this process to establish a robust visual emotional contrast. Subsequently, we prompt Gemini-3.0-Pro to generate paragraphs describing neutral events. These paragraphs are paired with both image variants to form contrastive sets, allowing us to investigate how specific visual emotional contexts modulate the model's response to otherwise neu-

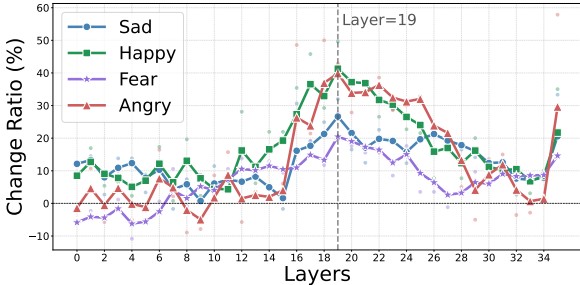

*(a)* Emotional Activation Steering on Qwen3-VL-4B-Instruct.

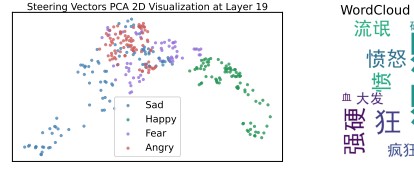
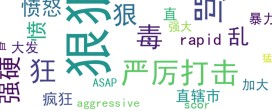

*(b)* PCA of steering vectors    *(c)* Wordcloud of Angry

*Figure 4.* Analyses of layer sensitivity and semantic projection demonstrate that critical emotional semantics aggregate and crystallize in the middle layers, serving as a pivotal stage.

tral textual events. The detailed processes of dataset construction and statistical checks are provided in Appendix B. **(2) Emotion Understanding Datasets** for validating the effectiveness of VEENA in mitigating emotional hallucinations. We adopt the comprehensive descriptive benchmark, MER-UniBench (Lian et al., 2025a), which covers diverse MER tasks including *Basic Emotion Recognition* (Lian et al., 2023; 2024b; Poria et al., 2019), *Sentiment Analysis* (Zadeh et al., 2017; 2018; Yu et al., 2020; Liu et al., 2022), and *Fine-grained Emotion Recognition* (Lian et al., 2024a). **(3) General Ability Evaluation Datasets** for assessing the robustness of our VEENA, which include POPE (Li et al., 2023), CHAIR (Rohrbach et al., 2018), and MME (Yin et al., 2024). The further details about these datasets are presented in Appendix A-B.

**Model Architecture.** Our primary analysis is centered on the current state-of-the-art and open-source LVLM from the Qwen series, *i.e.*, Qwen3-VL-4B-Instruct (Bai et al., 2025). To examine the impact of model scaling and architectural diversity, we extended our evaluation to include its 8B variant and LLaVA-OneVision-1.5-4B-Instruct (An et al., 2025).

**Implementation details.** We employ 250 contrastive pairs to extract the emotional directions across 4 basic emotions, and conduct mechanism analysis on another 250 (both randomly selected). We set the threshold $\mathcal{T} = 0.5$, intervention strength $\alpha = 0.1$, critical layer $l_{emo} = 19$, enhancement coefficient $\beta = 2.0$ and excitation coefficient $\gamma = 1.5$.

### 4.2. Emotional Circuit Analysis

In this section, we investigate (1) *when* critical semantics for coherent emotion portrayal emerge, (2) *where* this infor-

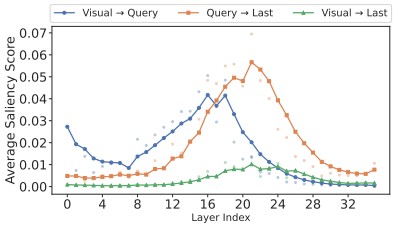 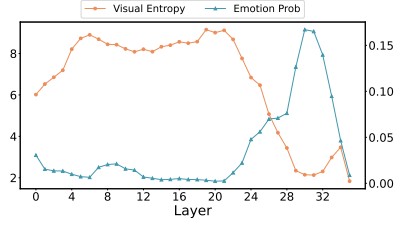 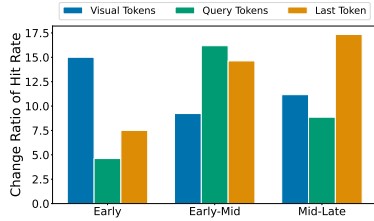

*(a)* Gradient-based Information Flow   *(b)* Visual Semantic Evolution   *(c)* Phase-level Activation Patching

*Figure 5.* Elucidating the Cross-Modal Emotion Routing Mechanism. **(a)** Visualizes the information flow dynamics across layers and modalities via sailency. **(b)** Traces the semantic entropy and emotion probability of visual tokens projected into the vocabulary space via Logit Lens. **(c)** Reports the causal impact on Emotion Hit Rate by patching activations of $V$, $Q$, and $L$ across distinct layer stages.

mation is integrated across modalities, and finally (3) *how* LVLMs extract and utilize these emotional cues.

### 4.2.1. LAYER-WISE EMOTION SEMANTICS EVOLUTION.

**Middle Layers as Emotion Pivots.** To map the trajectory of emotional semantic formation, we perform layer-wise interventions using the extracted emotion directions and monitor the model's behavior change. Notably, the change ratio exhibits a distinct distribution (Figure 4a): significant improvements are observed primarily in the middle layers ($l_{16}$-$l_{22}$), while early layers show negligible sensitivity. Notably, $l_{19}$ emerges as a critical inflection point, achieving the highest impact across multiple emotion categories (*e.g.*, $\approx 40\%$ in *Angry*) with divergent emotional latent difference groups (Figure 4b). This suggests that critical semantics for coherent emotion portrayal have emerged at this stage.

**Prior Excitation of Emotional Intents.** To decipher the representational content of the critical layers, we project the steering vector $S_{19}$ onto the vocabulary space via the unembedding matrix, resulting in human-readable concepts. As illustrated in Figure 4c, we observe a phenomenon of *Early Excitation*: concepts semantically aligned with the target emotion (*e.g.*, "aggressive" for *Angry*) exhibit peak activation probabilities precisely at this layer, significantly prior to the final output layer. This observation provides strong evidence that the middle layers serve as a pivotal stage for semantic aggregation, where the model crystallizes abstract emotional intents. Consequently, this finding justifies our focus on these specific layers for subsequent exploration.

### 4.2.2. CROSS-MODAL EMOTION ROUTING DYNAMICS.

To elucidate the mechanisms governing the extraction and utilization of emotional intents, we analyze the information flow across distinct token positions: the *Visual* ($V$), *Last* ($L$), and *Query* ($Q$, exclude $L$) tokens.

**Hypothesis Motivated by Saliency Scores.** We first employ a gradient-based saliency analysis (Wang et al., 2023), defined as the product of attention weights and the gradient of our $\mathcal{R}(\cdot)$ metric (see Section 3.2), to quantify the con-

tribution of information flow. As shown in Figure 5, the routing dynamics exhibit a distinct three-stage transition:

- *Latent Modality Adaptation* ($l_0$-$l_7$). while $V \rightarrow Q$ presents high saliency in this stage (Figure 5a), we observe a counter-intuitive "dip" with high visual token entropy (Figure 5b). We assume this phase represents a realignment where visual tokens transform from pixel-level encodings to LLM-compatible semantic spaces, rendering direct cross-modal attention temporarily ineffective.
- *Emotion Intents Aggregation* ($l_8$-$l_{18}$). Following adaptation, $V \rightarrow Q$ saliency rises sharply. Here, the Query tokens aggregate emotional contexts based upon visual stimuli to translate physical objects into abstract intents.
- *Emotional Generation Execution* ($l_{19}$-$l_{35}$). A structural shift occurs at $l_{19}$, where the dominance transfers from $V \rightarrow Q$ to $Q \rightarrow L$. The $L$ primarily relies on the emotionally enriched $Q$ for global tone, while retaining a secondary visual connection ($l_{14}$-$l_{21}$) to incorporate visual details.

**Verification via Causal Activation Patching.** To validate the hypothesized "Adapt-Aggregate-Execute" pathway, we conduct layer-wise activation patching on the $V$, $Q$, and $L$. By measuring the causal impact on the emotional hit rate, we reveal a causal landscape that strictly mirrors our saliency-based observations (Figure 5c): (1) *Adaptation* ($l_0$-$l_7$): Patching $V$ yields the most significant gain ($+15.0\%$). This confirms that early layers function as a foundational stage for aligning raw visual encodings. (2) *Aggregation* ($l_8$-$l_{18}$): The causal epicenter shifts to $Q$, which achieves the peak improvement ($+16.2\%$). This aligns with the aggregation phase, where the instruction actively absorbs emotional contexts from the adapted visual features. (3) *Execution* ($l_{19}$-$l_{35}$): Dominance transfers definitively to $L$ ($+17.3\%$), while the influence of $Q$ notably wanes. This indicates that once the emotional intent is crystallized, the model pivots to generation, leveraging the $Q$ to guide the decoding process. Collectively, these distinct causal profiles corroborate the three-stage emotional mechanism. Collectively, these distinct causal profiles corroborate the three-stage emotional mechanism. We further discuss the fine-grained emotion-level analysis in Appendix E.

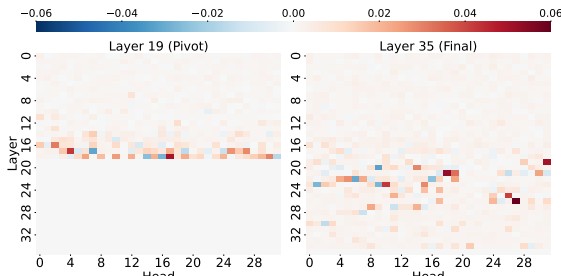

*(a)* Activation Patching of *Sad* from $l_{19}$ (left) and $l_{35}$ (right).

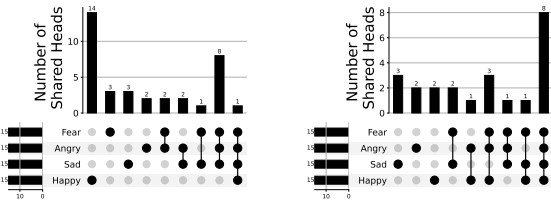

*(b)* Head Intersection at $l_{19}$     *(c)* Head Intersection at $l_{35}$

*Figure 6.* Functional Decoupling of Sentiment-Specific Aggregation and Universal Execution. **(a)** Visualizes causal heads for *Sad* at $l_{19}$ and $l_{35}$ layers. **(b-c)** Upset plots quantifying the intersection of causal heads across four basic emotions.

### 4.2.3. COMPOSITION OF EMOTIONAL CIRCUITS

To dissect the granular composition of the identified "Adapt-Aggregate-Execute" relay, we further scrutinize the attention heads and MLP neurons that drive these dynamics.

**Functional Decoupling of Specific Aggregation and Universal Execution.** We extend our circuit discovery to a dual-target setting, locating heads causal to emotional directions at both the pivot layer $l_{19}$ and the final layer $l_{35}$. As visualized in Figure 6a, we successfully identify critical upstream heads that are undetectable by traditional paradigms (see Section 2) restricted to the output layer, highlighting the superiority of our methods. Analyzing the intersection of these heads (Figure 6b-6c) reveals a distinct functional stratification: (1) *Upstream Sentiment-Specificity*: In "Aggregation", the intersection across all emotions is sparse; however, we observe cohesive clusters shared exclusively among negative sentiments (*i.e.*, *Sad*, *Angry*, and *Fear*). This indicates that the model dedicates specialized circuits to process distinct sentiment groups. (2) *Downstream Emotion-Agnosticism*: In contrast, heads in "Execution" demonstrate broad universality across the entire emotional spectrum. Consequently, these results validate a *Specific-to-Universal* processing hierarchy: the model requires specialized mechanisms to digest diverse emotional contexts but converges onto a unified pathway to final generation.

**Function Verification of Heads.** We conduct *Recovery* (from neutral) and *Knockout* (destruct emotional) experiments on the *Angry*. Figure 7 reveals that masking these critical heads precipitates a significant deterioration in emotional alignment, whereas their restoration suffices to re-

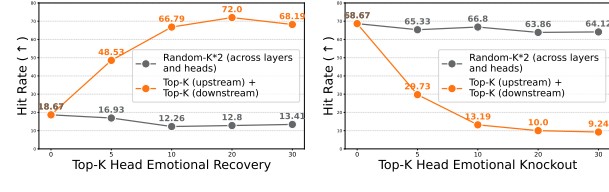

*(a)* Recovery (Patching)     *(b)* Knockout (Zero ablation)

*Figure 7.* Function Verification of the identified emotion-related heads. Notably, we sum the heads of both up- and downstream.

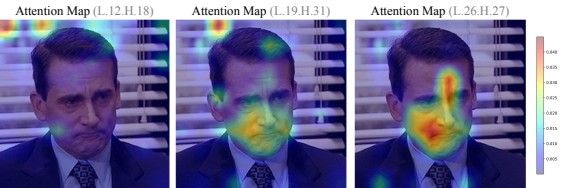

*Figure 8.* Attention-map visualization of universal visual head.

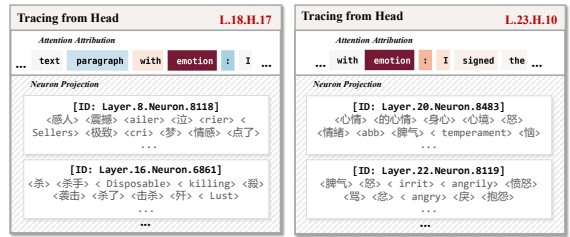

*Figure 9.* Emotional Neurons Visualizations of *Angry*.

cover the model's capability. In contrast, random heads of equal magnitude result in negligible performance fluctuations. These results confirm that the located heads play a crucial role in emotion processing.

**Universal Visual Grounding with Dynamic Focus.** We further analyze these heads exhibiting universality across diverse emotions, predominantly attending to visual regions. Visualizing their spatial patterns reveals a *Coarse-to-Fine* evolution (Figure 8): the focus shifts from global scopes for extracting holistic emotional cues to specific landmarks for grounding fine-grained details during *Execution*.

**Emotion-related Knowledge Storage.** Recursively tracing top-activating neurons reveals a distinct semantic dichotomy despite shared attention on the query anchor (Figure 9). Specifically, neurons in *Aggregation* encode *causal intents* (*e.g.*, "killing") as abstract precursors rather than the emotion itself. In contrast, *Execution* neurons crystallize these into definitive *emotional states* (*e.g.*, "angry"). This evolution from context-aware intents to explicit labels provides microscopic evidence validating the proposed *Aggregate-to-Execute* hierarchy.

### 4.3. Mitigating Emotional Hallucinations

**Effectiveness and Generalization.** We evaluated our VEENA on the comprehensive descriptive MER-uniBench.

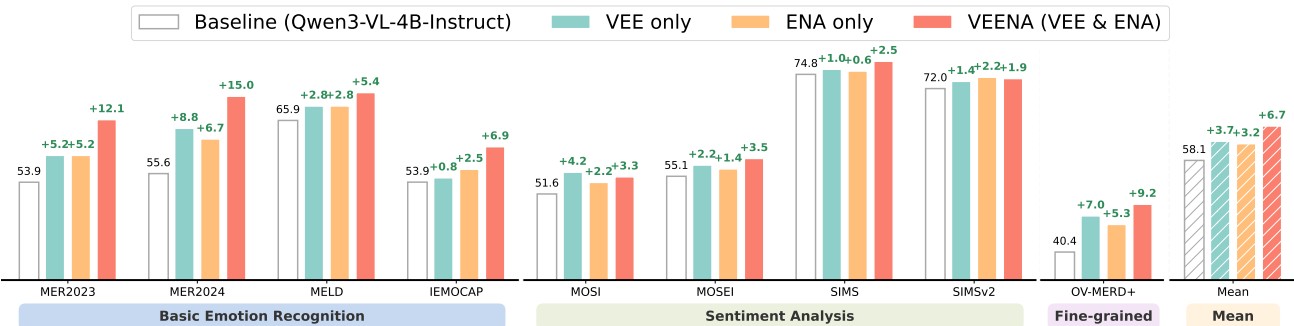

*Figure 10.* Experiment results on MER-uniBench average 3 seeds. We report the hit rate metric for Basic Emotion Recognition tasks, the weighted average F-score (WAF) for Sentiment Analysis tasks, and the $F_s$ for Fine-grained Emotion Recognition tasks. The details of these metrics and additional results of other models are presented in Appendix A.1 and Appendix C, respectively.

*Table 1.* Ablation of the selection scale for emotion-related heads and neurons. "Random": randomly select these components.

| Precise Setting | Selection Scale | Latency (ms/Token ↓) | MER2023 (hit rate ↑) | MOSI (WAF ↑) | OV-MERD+ ($F_s$ ↑) |
|---|---|---|---|---|---|
| *Heads for VEE-only (both upstream and downstream).* | | | | | |
| Top-5 | 1.82% | 14.73 | 58.54 | 54.15 | 43.09 |
| Top-10 | 3.56% | 14.78 | **59.12** | 55.86 | **47.42** |
| Top-20 | 6.25% | 14.91 | 58.98 | 55.27 | 43.36 |
| Top-30 | 9.46% | 14.96 | 57.56 | 54.52 | 47.22 |
| Random-10 | 3.56% | 14.78 | 56.35 | 53.77 | 41.93 |
| *Neurons for ENA-only Tracing From Top-10 Head* | | | | | |
| Top-10 | 1.25% | 14.72 | 53.14 | 52.70 | 39.38 |
| Top-20 | 2.32% | 14.74 | 56.05 | **54.16** | 40.51 |
| Top-30 | 3.36% | 14.75 | **59.12** | 53.89 | **45.76** |
| Top-40 | 4.30% | 14.75 | 57.23 | 51.64 | 40.42 |
| Random-30 | 3.36% | 14.75 | 54.99 | 49.81 | 41.11 |

*Table 2.* Comparison with previous training-free intervention methods on Qwen3-VL-4B-Instruct .

| Methods | MER2023 (Hit Rate ↑) | CMU-MOSI (WAF ↑) | OV-MERD+ ($F_s$ ↑) |
|---|---|---|---|
| Baseline | 53.92 | 51.65 | 40.42 |
| + VISTA (Li et al., 2025b) | 52.94 | 53.84 | 40.82 |
| + PAI (Liu et al., 2024b) | 59.61 | 54.37 | 46.32 |
| + VEENA (Ours) | **66.03** | **54.97** | **49.67** |

*Table 3.* Experiment results on the general benchmarks. The details of these metrics and complete results are presented in Appendix A.1 and Appendix D, respectively.

| Instruct Model | POPE | | CHAIR | | MME |
|---|---|---|---|---|---|
| | Acc ↑ | F1 ↑ | $C_S$ ↓ | $C_I$ ↓ | Score ↑ |
| Qwen3-VL-4B | 87.38 | 86.62 | 53.2 | 10.7 | 1661.1 |
| + VEENA | 87.62 | 87.04 | 51.0 | 10.3 | 1674.5 |
| | +0.24 | +0.42 | −2.2 | −0.4 | +13.4 |
| Qwen3-VL-8B | 87.90 | 87.37 | 51.8 | 10.1 | 1704.3 |
| + VEENA | 87.88 | 87.45 | 48.4 | 10.2 | 1706.8 |
| | −0.02 | +0.08 | −3.4 | +0.1 | +2.5 |
| LLAVA-OV-1.5-4B | 88.54 | 87.99 | 30.0 | 6.6 | 1574.4 |
| + VEENA | 88.33 | 88.17 | 31.4 | 7.4 | 1541.3 |
| | −0.21 | +0.18 | +1.4 | +0.8 | −33.1 |

Obviously, VEENA yields substantial improvements across all 9 datasets (Figure 10), significantly raising the average performance from 58.1% to 64.8%(+6.7%). Notably, VEENA outperforms standalone *VEE* and *ENA*, confirming the synergy of jointly regulating head routing and amplifying neuron knowledge for emotional understanding. For the most challenging OV-MERD+, VEENA achieves a remarkable gain (+9.2%). Crucially, this enhancement stems from genuine comprehension rather than a superficial bias to over-generate emotional vocabulary, as confirmed by our safety profiling in Appendix F. These results validate the causal fidelity of our identified circuits and establish surgical intervention as a highly efficient method without retraining. Still, we believe that precisely fine-tuning the emotional circuits holds significant potential.

**Comparison with Previous Methods.** To further demonstrate the advantages of our design, we supplemented our experiments on Qwen3-VL-4B-Instruct by comparing VEENA against recent SOTA inference-time methods: VISTA (Li et al., 2025b) and PAI (Liu et al., 2024b). As shown in Table 2, VEENA consistently outperforms these baselines across diverse MER tasks. We attribute this superiority to the fact that VEENA is explicitly grounded in our discovered emotion mechanism, providing a precise and interpretable intervention rather than a generalized adjustment.

**Ablation Study.** We further conducted ablation experiments

on the scale of selected heads and neurons. As shown in Table 1, the performances achieve *state-of-the-art* with intervention on ≈ 3% parameters, peaking at the Top-10 heads (both upstream and downstream) of each emotion and Top-30 neurons tracing from each head. This suggests that a concise set of critical components is sufficient to amplify emotional semantics. In contrast, random intervention fails to reproduce the significant improvement.

**Specificity of Emotional Circuits.** To further corroborate the functional disentanglement between the identified emotional circuits and general visual capabilities, we evaluated VEENA on standard general ability evaluation benchmarks (see Section 4.1). As reported in Table 3, VEENA does not yield universal improvements and even incurs slight trade-offs in certain metrics. These results demonstrate that VEENA successfully excites the *Emotion-Sub-Network* without indiscriminately over-activating basic object recognition mechanisms, thereby proving the orthogonality of our discovered circuits against general object perception.

## 5. Related Work

**Emotion Large Vision-Language Models.** Driven by the rapid advancement of LVLMs, a growing body of research investigates their potential for emotion understanding. EmoVIT (Xie et al., 2024) enhances the affective capabilities of InstructBLIP (Dai et al., 2023) by employing a GPT-assisted pipeline to generate diverse emotion-specific instruction data. Emotion-LLaMA (Cheng et al., 2024) integrates audio, visual, and textual features within a LLaMA-based framework to capture complex multimodal interactions. Similarly, AffectGPT (Lian et al., 2025a) employs a pre-fusion architecture to effectively integrate multimodal cues. Furthermore, AffectGPT-R1 (Lian et al., 2025b) and R1-Omni (Zhao et al., 2025) introduce reinforcement learning (Shao et al., 2024; Rafailov et al., 2023; Schulman et al., 2017) frameworks to significantly boost both accuracy and out-of-distribution robustness in emotion understanding. Although effective, these efforts frequently lead to catastrophic forgetting, thereby compromising general capabilities (Huang et al., 2025). In contrast, we aim to elucidate the internal mechanisms underlying emotion understanding in LVLMs, thereby enhancing emotional intelligence.

**Mechanism Interpretability.** Mechanistic interpretability (Madsen et al., 2023; Räuker et al., 2023; Geiger et al., 2025; Miao & Kan, 2025; Mondorf et al., 2025; Sun et al., 2025; 2026) represents an emerging frontier in NLP, aiming to reverse-engineer the granular computational mechanisms within neural networks. Recent studies (Jiang et al., 2024; Neo et al., 2024; Zhang et al., 2025; Kaduri et al., 2025; Luo et al., 2024) have scrutinized the internal mechanisms of LVLMs, specifically investigating how visual and textual modalities are integrated to drive linguistic generation. To identify key components of LVLMs in varied tasks, some studies have used causal analysis (Pearl, 2022), including Activation Patching (Basu et al., 2024; Golovanevsky et al., 2024; Li et al., 2025a) and Attribution Patching (Nikankin et al., 2025). However, existing studies have primarily focused on how LVLMs perceive concrete object entities. In contrast, we advance this line of inquiry by probing the internal mechanisms underlying the comprehension of abstract emotion.

## 6. Conclusion

This work pioneers a steering-vector-based attribution framework to demystify the emotion processing in LVLMs, revealing a distinct "Adapt-Aggregate-Execute" hierarchical mechanism. Our granular analysis uncovers a functional decoupling, where the model transitions from sentiment-specific intent aggregation with coarse visual cues to universal emotional execution grounded in fine-grained details. Guided by these insights, we introduce VEENA, a surgical, training-free intervention strategy that synergistically modulates information routing and amplifies semantic knowledge. Extensive experiments demonstrate that VEENA not only effectively mitigates emotional hallucinations—validating the causal fidelity of our mechanistic discoveries—but also robustly preserves general vision-language capabilities.

## Acknowledgements

This work was supported by the Natural Science Foundation of China under Grant 62571507.

## Impact Statement

Our research focuses strictly on elucidating the internal mechanistic pathways of existing open-source LVLMs to enhance their reliability and safety. The data utilized in this study are derived from established, publicly available academic benchmarks, ensuring no collection of private information or infringement of personal privacy. While the identification of emotional circuits theoretically introduces risks regarding affective manipulation, our proposed framework is explicitly engineered for *alignment*—mitigating hallucinations to ensure model responses accurately reflect visual reality rather than generating deceptive emotional content. Consequently, we aim to foster trustworthy human-AI interaction and improve the robustness of vision-language systems without incurring negative societal impacts.

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

*Table 4.* Statistics of the MER-UniBench dataset covering Fine-grained, Basic, and Sentiment Analysis tasks.

| Task Category | Dataset | Split | # Samples | Label Description | Data Source |
|---|---|---|---|---|---|
| **Basic** | MER2023 | MER-MULTI | 411 | Most likely label (6 candidates) | Movies, TV series |
| | MER2024 | MER-SEMI | 1,169 | Most likely label (6 candidates) | Movies, TV series |
| | IEMOCAP | Session5 | 1,241 | Most likely label (4 candidates) | Actor performance |
| | MELD | Test | 2,610 | Most likely label (7 candidates) | "Friends" TV series |
| **Sentiment Analysis** | CMU-MOSI | Test | 686 | Intensity score in $[-3, 3]$ | YouTube reviews |
| | CMU-MOSEI | Test | 4,659 | Intensity score in $[-3, 3]$ | YouTube reviews |
| | CH-SIMS | Test | 457 | Intensity score in $[-1, 1]$ | Movies, TV series |
| | CH-SIMS v2 | Test | 1,034 | Intensity score in $[-1, 1]$ | Movies, TV series |
| **Fine-grained** | OV-MERD+ | All | 532 | Unfixed categories & diverse labels | Movies, TV series |

## A. Experimental and Technical Details

### A.1. Metric Formula and Prompt Design

**Emotion Understanding Datasets.** We utilize the MER-UniBench (Lian et al., 2025a) to evaluate LVLMs' emotional intelligence and validating the effectiveness of our VEENA in mitigating emotional hallucinations. As shown in Table 4, the MER-UniBench contains three distinct MER tasks:

- **Basic Emotion Recognition** including MER2023 (Lian et al., 2023), MER2024 (Lian et al., 2024b), MELD (Poria et al., 2019), and IEMOCAP (Busso et al., 2008)) datasets. To bridge the gap between these fixed labels and the free-form responses of LVLMs, we adopt the *Hit Rate* metric ($\mathcal{H}$) defined in Section 2, which evaluates whether the ground-truth emotion is encompassed within the model's generated keywords after mapping to standardized emotion wheels.

- **Fine-grained Emotion Recognition** including OV-MERD+ (Lian et al., 2024a) datasets. For tasks requiring diverse emotional descriptors (Lian et al., 2024a), the model output $\hat{\mathbf{Y}}$ and ground truth $\mathbf{Y}$ are treated as variable-length sets. To mitigate linguistic ambiguity, we apply a hierarchical grouping function $G_{w_k}(\cdot)$ that unifies synonyms and maps outer-wheel nuances to inner core emotions across $K$ distinct emotion wheels. We report the $F_s$, which aggregates performance across all wheels:

$$F_s = \frac{1}{K} \sum_{k=1}^{K} \frac{2 \cdot \text{Precision}_s^k \cdot \text{Recall}_s^k}{\text{Precision}_s^k + \text{Recall}_s^k}, \quad \text{where Precision}_s^k = \frac{|G_{w_k}(\mathbf{Y}) \cap G_{w_k}(\hat{\mathbf{Y}})|}{|G_{w_k}(\hat{\mathbf{Y}})|}, \text{Recall}_s^k = \frac{|G_{w_k}(\mathbf{Y}) \cap G_{w_k}(\hat{\mathbf{Y}})|}{|G_{w_k}(\mathbf{Y})|},$$
(13)

- **Sentiment Analysis** including MOSI (Zadeh et al., 2017), MOSEI (Zadeh et al., 2018), SIMS (Yu et al., 2020), and SIMSv2 (Liu et al., 2022)) datasets. For datasets annotated with continuous sentiment intensity (*e.g.*, scores in $[-3, 3]$) (Zadeh et al., 2017; 2018), we perform polarity mapping by binarizing scores into *Positive* ($> 0$) and *Negative* ($< 0$) categories, excluding neutral samples. To address inherent label imbalances, we report the *Weighted Average F-score (WAF)* as the primary metric, complemented by standard Accuracy (ACC).

Each sample comprises multi-modal data (*e.g.*, audio, video, and caption) curated through a model-led, human-assisted pipeline. In our experiments, we focus specifically on the video modality, extracting the middle frame from each video to serve as the image input. Following AffectGPT (Lian et al., 2025a), we prompt the model for responses as:

> **[System Prompt]**
> You are an expert in the field of emotions.
> **[Task Prompt]**
> Please focus on the facial expressions, body movements, environment, subtitle content, etc., in the image to discern clues related to the emotions of the individual. Please provide a detailed description and ultimately predict the emotional state of the individual in the image:

**General Ability Evaluation Datasets.** To ensure that our interventions do not compromise the foundational capabilities of LVLMs, we conduct evaluations on three widely adopted benchmarks:

*Table 5.* Experiment results of **Qwen3-VL-8B-Instruct** on MER-uniBench average 3 seeds.

| Method | Basic Emotion Recognition | | | | Sentiment Analysis | | | | Fine-grained | Mean |
|---|---|---|---|---|---|---|---|---|---|---|
| | MER2023 | MER2024 | MELD | IEMOCAP | MOSI | MOSEI | SIMS | SIMSv2 | OV-MERD+ | |
| Qwen3-VL-8B | 60.68 | 62.49 | 67.70 | 55.61 | 54.41 | 56.79 | 77.03 | 76.48 | 47.04 | 62.03 |
| EVE-only | 62.82 | 66.48 | 70.27 | 57.13 | 54.76 | 57.53 | 77.17 | 76.95 | 49.09 | 63.58 |
| | +2.14 | +3.99 | +2.57 | +1.52 | +0.35 | +0.74 | +0.14 | +0.47 | +2.05 | +1.55 |
| ENA-only | 61.90 | 64.91 | 68.68 | 55.05 | 53.84 | 56.51 | 77.82 | 76.85 | 47.79 | 62.59 |
| | +1.22 | +2.42 | +0.98 | −0.56 | −0.57 | −0.28 | +0.79 | +0.37 | +0.75 | +0.56 |
| VEENA | 63.46 | 70.03 | 71.18 | 55.97 | 54.52 | 57.99 | 77.59 | 77.62 | 49.86 | 64.25 |
| | +2.78 | +7.54 | +3.48 | +0.36 | +0.11 | +1.20 | +0.56 | +1.14 | +2.82 | +2.22 |

*Table 6.* Experiment results of **LLaVA-OneVision-1.5-4B-Instruct** on MER-uniBench average 3 seeds.

| Method | Basic Emotion Recognition | | | | Sentiment Analysis | | | | Fine-grained | Mean |
|---|---|---|---|---|---|---|---|---|---|---|
| | MER2023 | MER2024 | MELD | IEMOCAP | MOSI | MOSEI | SIMS | SIMSv2 | OV-MERD+ | |
| LLAVA-OV-1.5-4B | 48.52 | 56.75 | 63.80 | 53.55 | 50.04 | 52.88 | 68.73 | 66.22 | 42.93 | 55.94 |
| EVE-only | 53.63 | 62.28 | 65.49 | 56.26 | 52.75 | 55.55 | 71.82 | 70.73 | 44.89 | 59.27 |
| | +5.11 | +5.53 | +1.69 | +2.71 | +2.71 | +2.67 | +3.09 | +4.51 | +1.96 | +3.33 |
| ENA-only | 51.48 | 58.49 | 64.11 | 53.84 | 50.81 | 52.10 | 70.29 | 64.91 | 43.44 | 56.61 |
| | +2.96 | +1.74 | +0.31 | +0.29 | +0.77 | −0.78 | +1.56 | −1.31 | +0.51 | +0.67 |
| VEENA | 55.04 | 62.80 | 66.07 | 56.95 | 53.51 | 54.91 | 70.98 | 70.98 | 45.37 | 59.62 |
| | +6.52 | +6.05 | +2.27 | +3.40 | +3.47 | +2.03 | +2.25 | +4.76 | +2.44 | +3.68 |

- **POPE** (Li et al., 2023) quantitatively assesses object hallucination via a binary classification task ("`Yes`"/"`No`"). The benchmark queries the existence of objects using the template "`Is there a <object> in the image?`", categorized into three splits of escalating difficulty: *Random*, *Popular*, and *Adversarial*. Each split consists of $3,000$ VQA pairs sampled from the MSCOCO (Lin et al., 2014) validation set. We report performance using both Accuracy (ACC) and F1-score (F1).

- **CHAIR** (Rohrbach et al., 2018) measures object hallucination in open-ended generation tasks. Following prior work (Li et al., 2025b), we randomly sample $500$ images from the MSCOCO validation set and query the model with "`Please help me describe this image in detail.`". We compute two metrics: instance-level ($\text{CHAIR}_I$) and sentence-level ($\text{CHAIR}_S$), defined as:

$$\text{CHAIR}_I = \frac{|\{\text{hallucinated objects}\}|}{|\{\text{total objects mentioned}\}|}, \quad \text{CHAIR}_S = \frac{|\{\text{captions w/ hallucination}\}|}{|\{\text{total captions}\}|}. \tag{14}$$

- **MME** (Yin et al., 2024) is employed to holistically evaluate the model's perception capabilities. We specifically utilize the perception suite, which encompasses 10 diverse subtasks (including coarse-grained and fine-grained recognition). We report the aggregate accuracy score across these perception tasks to gauge overall robustness.

### A.2. Implementation Details and Reproducibility

**Model Configuration.** We implement all experiments using the PyTorch framework. To balance computational efficiency with numerical stability, all LVLMs are loaded in `bfloat16` precision. Crucially, for our mechanistic interpretability analysis (*e.g.*, activation patching and circuit discovery), we explicitly set the attention implementation to "`eager`" rather than Flash Attention (Dao et al., 2022).

**Inference Hyperparameters.** To ensure the determinism and reproducibility of our results, we adhere to a strict greedy decoding strategy with a fixed repetition penalty at $1.0$ across all evaluation benchmarks. The generation length is constrained with a minimum of 32 tokens to avoid truncated outputs and a maximum of $512$ tokens to prevent infinite loops. We enable KV-caching to accelerate the inference process during the token generation phase.

### A.3. Compute Resources

All our experiments were executed on a computing node equipped with $4$ NVIDIA RTX 4090 GPUs, each possessing 24 GB of memory. Peak memory utilization was observed during the emotional neuron discovery phase for Qwen3-VL-8B-Instruct,

Table 7. Full Generalization Results on **POPE**. Performance is measured by accuracy (%) and F1-score (%).

| Instruct Model | Random | | Popular | | Adversarial | | Avg | |
|---|---|---|---|---|---|---|---|---|
| | Accuracy ↑ | F1-score ↑ | Accuracy ↑ | F1-score ↑ | Accuracy ↑ | F1-score ↑ | Accuracy ↑ | F1-score ↑ |
| Qwen3-VL-4B | 89.28 | 88.60 | 87.07 | 86.20 | 85.80 | 85.07 | 87.38 | 86.62 |
| + VEENA | 89.83 | 89.28 | 87.33 | 86.67 | 85.70 | 85.20 | 87.62 | 87.04 |
| | +0.55 | +0.68 | +0.27 | +0.46 | -0.10 | +0.13 | +0.24 | +0.42 |
| Qwen3-VL-8B | 90.07 | 89.57 | 87.97 | 87.30 | 85.67 | 85.23 | 87.90 | 87.37 |
| + VEENA | 90.41 | 89.97 | 87.87 | 87.31 | 85.37 | 85.06 | 87.88 | 87.45 |
| | +0.34 | +0.40 | -0.10 | +0.01 | -0.30 | -0.17 | -0.02 | +0.08 |
| LLAVA-OV-1.5-4B | 89.90 | 89.38 | 88.97 | 88.31 | 86.77 | 86.29 | 88.54 | 87.99 |
| + VEENA | 90.52 | 90.26 | 88.90 | 88.59 | 85.57 | 85.65 | 88.33 | 88.17 |
| | +0.62 | +0.88 | -0.07 | +0.28 | -1.20 | -0.64 | -0.21 | +0.18 |

Table 8. Full Generalization Results on **MME**. Performance is measured by the accuracy counts.

| Method | MME Perception Tasks | | | | | | | | | | Score |
|---|---|---|---|---|---|---|---|---|---|---|---|
| | Existence | Count | Position | Color | Posters | Celebrity | Scene | Landmark | Artwork | OCR | |
| Qwen3-VL-4B | 190.0 | 171.7 | 153.3 | 188.3 | 164.3 | 163.8 | 149.0 | 148.8 | 139.5 | 192.5 | 1661.1 |
| + VEENA | 190.0 | 173.3 | 158.3 | 188.3 | 161.9 | 160.9 | 158.8 | 158.3 | 139.8 | 185.0 | 1674.5 |
| | − | +1.7 | +5.0 | − | −2.4 | −2.9 | +9.8 | +9.5 | +0.3 | −7.5 | +13.4 |
| Qwen3-VL-8B | 200.0 | 156.7 | 146.7 | 190.0 | 174.8 | 180.6 | 148.5 | 171.5 | 150.5 | 185.0 | 1704.3 |
| + VEENA | 200.0 | 156.7 | 146.7 | 190.0 | 174.1 | 182.4 | 150.0 | 172.3 | 149.8 | 185.0 | 1706.8 |
| | − | − | − | − | −0.7 | +1.8 | +1.5 | +0.8 | −0.8 | − | +2.5 |
| LLAVA-OV-1.5-4B | 200.0 | 155.0 | 146.7 | 180.0 | 165.6 | 140.3 | 163.3 | 171.3 | 134.8 | 117.5 | 1574.4 |
| + VEENA | 200.0 | 155.0 | 136.7 | 175.0 | 163.6 | 133.2 | 167.0 | 164.5 | 136.3 | 110.0 | 1541.3 |
| | − | − | −10.0 | −5.0 | −2.0 | −7.1 | +3.8 | −6.8 | +1.5 | −7.5 | −33.1 |

necessitating the parallel deployment of 2 GPUs. The total computational cost for the comprehensive experimental suite, including preliminary trials, amounted to approximately $\approx 250$ GPU hours.

## B. Mechanistic Analysis Dataset Description

### B.1. Emotional Image Collection and Screening Protocol

To construct a rigorous foundation for mechanistic discovery, we aggregate a corpus of affect-rich images from publicly accessible web sources. To preclude potential confounders and ensure high signal fidelity, we enforce four rigorous inclusion criteria:

- **Visual Purity.** We strictly exclude images containing typographic elements, such as subtitles, watermarks, or embedded symbols. This constraint prevents the model from exploiting "OCR shortcuts" (*i.e.*, reading text labels) rather than processing intrinsic visual emotional cues.

- **Subject Diversity.** The dataset spans a broad spectrum of domains, including human portraits, animal behaviors, anime characters, and abstract emojis. This diversity ensures that the identified circuits respond to the abstract semantics of "emotion" rather than overfitting to domain-specific features (*e.g.*, focusing solely on human facial landmarks).

- **Emotional Singularity.** Each sample is verified to exhibit a single, dominant emotional state, effectively eliminating samples with ambiguous, subtle, or mixed sentiments that could confound the specific circuit tracing process.

- **Stimulus Validity.** To guarantee the efficacy of the visual stimuli, we subject the candidate images to a pre-screening classification task using the target LVLM. Only samples where the model predicts the ground-truth emotion with high confidence ($> 95\%$ accuracy) are retained. This ensures that the selected images act as robust stimuli capable of effectively activating the target internal emotional pathways.

### B.2. Emotion Contrast Construction

To rigorously isolate emotional circuits from general visual or linguistic processing, we construct strict counterfactual pairs using an automatic generative pipeline. This process ensures that the resulting dataset, $\mathcal{D} = \{(I_{emo}, T), (I_{neu}, T)\}$, differs

solely in the visual affect variable while holding all other semantic factors constant. For each screened emotional image $I_{emo}$, we employ Google Nano Banana to generate a paired neutral counterfactual $I_{neu}$. To prevent the introduction of confounding variables, we designed a prompt with *Immutable Constraints* and *Targeted Neutralization* protocols as shown in Figure 12. The textual component $T$ consists of "Neutral Events" designed to validate the model's ability to interpret context. Drawing on the *Kuleshov Effect* (Prince & Hensley, 1992; Mobbs et al., 2006) theory, we prompt Gemini-3.0-Pro to generate descriptions that satisfy three criteria as shown in Figure 13.

### B.3. Quantitative Verification of Counterfactual Consistency

Isolating the "emotion variable" is paramount for rigorous mechanistic analysis. To guarantee the "pure neutralization" of our counterfactual pairs and verify that the generative pipeline introduces no subtle semantic, texture, or identity shifts (which could act as uncontrollable confounders during circuit localization), we conduct comprehensive quantitative validations across three dimensions:

- **Global Visual Semantic Consistency.** We compute the cosine similarities of global image embeddings extracted via the vision encoders of our target LVLMs. As reported in Table 9, our $(I_{emo}, I_{neu})$ pairs maintain near-perfect visual consistency (similarity $> 0.97$), whereas random pairings exhibit no meaningful correlation. This confirms that the overall visual semantics are strictly preserved during the neutralization process.

*Table 9.* Global semantic consistency measured by cosine similarity of LVLM vision encoder embeddings.

| Evaluation Set | Qwen3-VL-4B-Instruct | LLaVA-OV-1.5-4B-Instruct | Qwen3-VL-8B-Instruct |
|---|---|---|---|
| $I_{emo}$ vs $I_{neu}$ | 0.9742 | 0.9747 | 0.9742 |
| Random Pairs | $-0.1671$ | $-0.7429$ | $-0.1671$ |

- **Facial Identity Preservation.** For the subset of images containing human subjects, we evaluate identity retention using the DeepFace framework (Taigman et al., 2014). We measure identity similarities across four expert facial recognition models, including VGG-Face (Parkhi et al., 2015), ArcFace (Deng et al., 2019), GhostFaceNet (Alansari et al., 2023), and FaceNet512 (Schroff et al., 2015). As shown in Table 10, the similarities of our counterfactual pairs significantly outperform random baselines by a large margin, demonstrating that core facial features and identity details are robustly retained despite the neutralized expressions.

*Table 10.* Facial identity preservation evaluated across expert facial recognition models.

| Evaluation Set | VGG-Face | ArcFace | GhostFaceNet | Facenet512 |
|---|---|---|---|---|
| $I_{emo}$ vs $I_{neu}$ | 0.4540 | 0.5317 | 0.3373 | 0.4943 |
| Random Pairs | 0.0661 | 0.0351 | 0.0668 | 0.1951 |

- **Consistency in Non-Emotional Tasks.** To further prove that the pairings remain unchanged in non-affective dimensions, we prompt the LVLMs with visual consistency verification tasks focusing on main objects, background attributes, and physical identity traits (explicitly instructing the models to ignore facial expressions). Table 11 illustrates that the models identify the $I_{emo}$ and $I_{neu}$ images as identical across these non-emotional axes with near-perfect accuracy ($> 96\%$).

*Table 11.* Consistency accuracy (%) in non-emotional Visual QA tasks.

| Consistency Type | Qwen3-VL-4B-Instruct | LLaVA-OV-1.5-4B-Instruct | Qwen3-VL-8B-Instruct |
|---|---|---|---|
| Object | 98.87 | 100.0 | 96.59 |
| Attribute (Background) | 96.59 | 98.87 | 98.87 |
| Identity (Traits/Age) | 97.72 | 100.0 | 100.0 |

### B.4. Task Formulation

Leveraging the constructed contrastive pairs, we design a *Emotion Continuation Task* to probe the causal mechanisms of emotional processing. In this paradigm, the model is presented with the same neutral textual event $T$ paired with either an

emotional image $I_{emo}$ or its neutral counterfactual $I_{neu}$. The objective is to investigate how specific visual affective cues actively modulate the model's interpretation of the ambiguous text, shifting the narrative tone from a baseline neutral state to a distinct emotional trajectory. By analyzing the divergence in internal activations and generation outputs between these paired inputs, we can effectively isolate the specific circuits responsible for integrating emotion into linguistic reasoning.

---

**[System]**
Imagine you are a real person rather than an assistant.
**[Task]**
Please continue writing the following text paragraph with emotion:

---

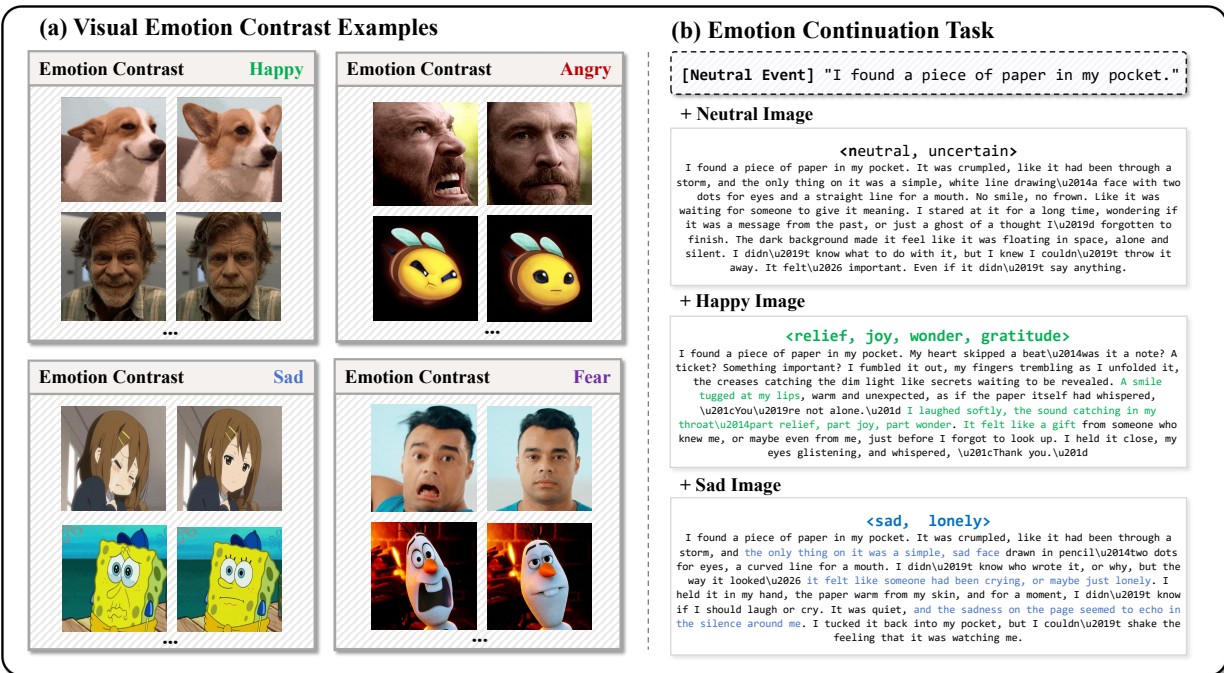

*Figure 11.* Case study. **(a)** Our mechanistic analysis dataset features diverse image pairs across multiple domains (photorealistic, anime, abstract), each consisting of an emotional stimulus and a strictly aligned neutral counterfactual. **(b)** A fixed neutral textual event produces distinct emotional narratives (*e.g.*, shifting from *uncertainty* to *joy* or *sadness*) solely driven by the visual affective context.

## B.5. Case Study

To qualitatively introduce the Emotion Continuation Task, we analyze the model's generation (Qwen3-VL-4b-Instruct) dynamics under varying visual stimuli. As illustrated in Figure 11(b), given the neutral context "I found a piece of paper in my pocket," the narrative trajectory diverges significantly based on the injected visual affect. When paired with a *Neutral* image, the continuation remains ambiguous and introspective ("*waiting for someone to give it meaning*"). Conversely, the *Happy* stimulus steers the generation toward "*relief*" and "*gratitude*" ("*A smile tugged at my lips*"), while the *Sad* stimulus evokes themes of "*loneliness*" where the "*sadness... seemed to echo.*" This distinct semantic shift confirms that the constructed counterfactual pairs successfully isolate and modulate the model's emotional reasoning capabilities without altering the textual premise.

## B.6. Dataset Release and Reproducibility

To foster transparency and accelerate future research, we commit to fully open-sourcing our assets upon paper acceptance. This release encompasses the our complete mechanistic analysis dataset—featuring paired pixel-perfect neutral counterfactuals and contextual text prompts—alongside the full codebase for our hierarchical circuit discovery pipeline and the VEENA intervention framework, strictly adhering to reproducible research principles to facilitate future mechanistic inquiries.

## C. Additional Experimental for Mitigating Emotional Hallucination

To assess the scalability and architectural universality of our intervention, we extended the evaluation to larger-scale and structurally distinct models, specifically Qwen3-VL-8B-Instruct and LLaVA-OneVision-1.5-4B-Instruct. As detailed in Table 5 and 6, VEENA consistently outperforms baselines across diverse settings, achieving substantial mean improvements of $+2.22\%$ and $+3.68\%$, respectively. Notably, while the neuron-centric ENA module exhibits slight volatility in specific Sentiment Analysis tasks compared to the robust attention-centric VEE, their synergistic combination within VEENA invariably yields the optimal performance. This evidence strongly validates that the identified "Adapt-Aggregate-Execute" mechanism is intrinsic to LVLMs, confirming that our surgical intervention offers a generalized solution for mitigating emotional hallucinations regardless of model size or architecture.

## D. Full Results for General Capability Preservation

This section provides the comprehensive, granular results of our general capability evaluation, supplementing the aggregated findings in Section 4.1. Table 7 details the performance across the *Random*, *Popular*, and *Adversarial* splits of the POPE benchmark, while Table 8 breaks down the scores for the 10 distinct perception subtasks within MME. These results consistently reveal that our VEENA maintains high stability across diverse visual perception categories. Although minor trade-offs are observed in specific fine-grained tasks (*e.g.*, OCR or Color) for LLaVA-OneVision-1.5-4B-Instruct, the overall performance—particularly in high-level semantics such as Scene and Landmark understanding—remains robust or even improves for Qwen3 variants. These detailed breakdowns further substantiate that our circuit-specific intervention operates orthogonally to the model's fundamental object recognition and reasoning mechanisms.

## E. Extrapolation to Fine-Grained Emotions

A critical consideration in affective computing is whether mechanisms discovered on basic emotions extrapolate to a more complex emotion spectrum. Humans express over 34,000 distinct emotions (Plutchik, 2001), making exhaustive granular labeling computationally prohibitive. To address this, we follow the methodology of AffectGPT (Lian et al., 2025a) to evaluate open-ended expressions by hierarchically mapping fine-grained generated emotions (*e.g.*, "hopeful", "loving") to core basic anchors (*e.g.*, "happy").

To explicitly verify whether our discovered functional decoupling extrapolates to these complex emotions, we conducted a phase-level activation patching experiment for the fine-grained emotion "hopeful". Instead of measuring the basic emotion hit rate, we strictly tracked the specific **generation frequency** of the target word "hopeful".

*Table 12.* Generation frequency of the fine-grained emotion word "hopeful" during phase-level activation patching, evaluated on Qwen3-VL-4B-Instruct.

| Token | Early Phase | Early-Mid Phase | Mid-Late Phase |
|---|---|---|---|
| Visual | **0.036** | 0.022 | 0.021 |
| Query | 0.025 | **0.027** | 0.032 |
| Last | 0.022 | 0.024 | **0.036** |

As demonstrated in Table 12, the causal trajectory perfectly replicates our "Adapt-Aggregate-Execute" mechanism. Specifically, the maximum activation smoothly transitions from the Visual token in the Early Phase, to the Query token in the Early-Mid Phase, and finally to the Last token in the Mid-Late Phase. This confirms that the functional decoupling of "middle-layer emotion-specific aggregation to deep-layer emotion-general execution" universally holds for fine-grained emotional subsets.

## F. Safety and Side-Effect Profile of VEENA

To systematically verify if the VEENA intervention induces "over-emotionalization" or emotional bias, we evaluated the Qwen3-VL-4B-Instruct model strictly on neutral images ($I_{neu}$) across two diagnostic tasks. We tested both greedy decoding and sampling decoding strategies (matching the hyperparameters used in the main evaluation) to observe if decoding styles amplify potential side effects.

- **Emotion Recognition on Neutral Images:** This task assesses whether the model can still correctly identify neutral states

without bias.

> **[Task Prompt]**
> Is this emotion happy, sad, angry, fear, neutral or else?
> **[Evaluation Metric]**
> Accuracy (%)

- **Emotion-Agnostic Image Description:** This task verifies whether the model simply generates more emotion-related vocabulary regardless of the context.

> **[Task Prompt]**
> Please describe this image in detail, focusing on the main subjects and their actions.
> **[Evaluation Metric]**
> LLM-as-Judge Emotion Score (Scale from 1: Completely objective to 5: Highly emotional

*Table 13.* Quantitative evaluation of VEENA's side-effect profile on neutral stimuli across different decoding strategies.

| Task | Metric | Greedy Decoding | Baseline | Baseline + VEENA |
|---|---|:---:|:---:|:---:|
| Recognition | Accuracy (%) ↑ | ✓ | 98.86 | 96.59 |
| | | ✗ | 90.90 | 88.63 |
| Description | Emotion Score ↓ | ✓ | 1.43 | 1.59 |
| | | ✗ | 1.61 | 1.69 |

As summarized in Table 13, the results demonstrate that VEENA does not trigger severe over-emotionalization. The accuracy drops in neutral image recognition are marginal ($\sim 2\%$) across both decoding strategies. Furthermore, the Emotion Scores during general image descriptions remain strictly objective, proving that the model does not hallucinate affective vocabulary when prompted for factual descriptions.

While the side effects are well-controlled under our fixed intervention thresholds, we acknowledge that implementing dynamic threshold modulation strategies based on prompt context could further optimize the safety profile. This serves as an encouraging avenue for deploying mechanistic interventions safely in practical scenarios.

## G. Limitations

While our framework establishes a foundational understanding of emotional mechanisms in LVLMs, several limitations delineate the scope of our current findings and point toward future avenues. First, our investigation is confined to static visual stimuli (*i.e.*, image); however, human emotional expression is inherently multimodal, encompassing temporal dynamics in video, prosody in audio, and textual nuances, which necessitates future extension to unified multimodal settings. Furthermore, constrained by computational resources, our analysis focuses on accessible models in the 4B to 8B parameter range, leaving it an open question whether emergent emotional circuits or distinct routing mechanisms manifest in significantly larger-scale models. Finally, we prioritized basic emotion categories to ensure high signal fidelity for mechanism discovery; consequently, our framework may not fully capture the complexity of mixed, subtle, or high-context affective states—such as sarcasm or bittersweetness—which likely involve more entangled and distributed representational architectures.

---

**Prompt Template for Counterfactual Image Generation**

```
Modify the image to strictly neutralize ALL emotional indicators while preserving the original composition, identity, and style with
pixel-level precision. The goal is to render the subject completely indifferent, calm, and expressionless.

STRICT CONSTRAINTS (Immutable Elements):
1.  **Structural Integrity:** Do NOT change the subject's identity, overall body posture, limb position, or silhouette.
2.  **Background Locking:** The background, scenery, and lighting must remain pixel-perfectly identical to the original.
3.  **Style Consistency:** Maintain the exact artistic medium (e.g., photo, oil painting, anime), color grading, and texture.

TARGETED NEUTRALIZATION (Apply only where necessary):

1.  **Facial Reset (The "Poker Face"):**
    * **Mouth:** Close the mouth to a neutral, resting line. Remove all smiles, frowns, sneers, or screams. Hide teeth and gums. Un-purse
lips.
    * **Eyes:** Restore eyelids to a normal, relaxed aperture (no wide-eyed fear, squinting, or crying). Ensure pupils are normal size.
    * **Brows:** Completely smooth the forehead. Remove furrowed (anger), raised (surprise), or slanted (sadness) eyebrows.
    * **Micro-expressions:** Eliminate any tension in the cheeks, jaw, or chin.

2.  **Physiological & Fluid Cleanup:**
    * **Fluids:** Erase ALL tears, sweat drops, saliva, drool, or nasal mucus.
    * **Skin Tone:** Remove emotional flushing (blush/redness) or pallor (pale fear).
    * **Veins:** Remove bulging veins associated with anger or stress.

3.  **Symbolic, Textual & Object Removal:**
    * **Symbols/Effects:** Erase emotional iconography (e.g., anime anger veins, depression gloom lines, sparkles, hearts, lightning
bolts).
    * **Text & Speech:** Remove any text, speech bubbles, or sound effect visualizations (e.g., "Ahhh!", "Sob") related to the emotion.
    * **Emotional Props:** Remove or neutralize objects that exist *solely* to convey emotion (e.g., remove a party hat, replace a tear-
soaked tissue with a neutral hand position, remove a weapon if it implies immediate threat/aggression, or make the grip relaxed).

4.  **Non-Human/Animal Specifics (If applicable):**
    * **Ears & Tail:** Reset ears to a standard forward/relaxed position (no pinned ears). Smooth out piloerection (puffed fur).
    * **Snout:** Smooth out any wrinkling or snarling on the nose.

**Result Vision:** A "Twin Image" where the subject looks bored, apathetic, and completely chemically balanced, distinct only by the total
absence of the original emotion.
```

*Figure 12.* Prompt for using Google Nano Banana to generate counterfactual images in emotional mechanistic analysis.

---

**Prompt Template for Neutral Event Generation**

```
# Role
You are an expert linguist and data generator specializing in Pragmatics and Multimodal Sentiment Analysis. Your task is to generate a
dataset of "Neutral Event."

# Definition
A "Neutral Event" is a short, factual description of an event or state that contains **zero inherent emotion** in the text itself. However,
the semantic meaning of the sentence must change drastically based on the emotion expression or tone accompanying it.

# The Kuleshov Effect Criteria
For each sentence you generate, it must satisfy the following conditions:
1. **Neutrality:** It must NOT contain emotionally charged adjectives or adverbs (e.g., avoid words like "finally," "unfortunately,"
"great," "disaster," "success").
2. **Concrete:**
   - It must describe a specific action, event, or observation.
   - It must contain more than 5 words.
   - It can use pronouns in the first person (e.g., 'me', 'my', and 'I'), but no pronouns in the third person  (e.g., 'she' and 'he') or
second person  (e.g., 'you' and 'your') are allowed.
3. **High Variance:**
   - If paired with a HAPPY/JOY/EXCITED emotion, it is interpreted as positive.
   - If paired with an ANGRY/SAD/DISGUST/FEAR emotion, it is interpreted as negative.
   - If paired without any emotion, the corresponding possible response should be neutral.

# Example
[
  {
    "sentence": "My mom said that we will be away for a few days.",
    "interpretation_positive": "We are going on a fun vacation or a surprise trip.",
    "interpretation_negative": "We are fleeing home due to a family emergency or danger."
  },
]

# Task
Generate new entries following the format and criteria above. Ensure the scenarios cover diverse domains (Office, Relationships, Medical,
Daily Chores, etc.). Do not repeat yourself.
"""
```

*Figure 13.* Prompt for using Gemini-3.0-Pro to generate neutral events in emotional mechanistic analysis.

