# OpenReview forum: "Interpreting and Enhancing Emotional Circuits in Large Vision-Language Models via Cross-Modal Information Flow"
_ICML.cc/2026/Conference — ICML 2026 regular_

### Official Review · Reviewer_UcKy · 2026-03-06

**Soundness:** 3
**Presentation:** 3
**Significance:** 3
**Originality:** 3
**Overall Recommendation:** 5
**Confidence:** 3

**Summary:**

Targeting the "emotional hallucination" issue frequently occurring in Large Vision-Language Models (LVLMs) during emotion understanding, this paper proposes a steering-vector-based causal attribution framework to parse cross-modal emotional information flows. By extracting emotional directions across layers using paired emotional and neutral counterfactual images, the study successfully uncovers a three-stage "Adapt-Aggregate-Execute" mechanism within the model. The findings reveal that the model completes modality alignment in the shallow layers, relies on specific attention heads to aggregate visual emotional intents into the Query token in the middle layers, and translates these into the final emotional expression for narrative generation through universal pathways in the deep layers. Based on this mechanism, the authors design a training-free, inference-time intervention framework called VEENA. This framework utilizes VEE to enhance the attention routing of emotional cues and combines ENA to amplify the emotional semantic activation of MLP neurons. Experiments demonstrate that VEENA significantly improves emotion recognition performance and effectively mitigates emotional hallucinations on the MER-UniBench, while its impact on general visual perception capabilities remains orthogonal and controllable.

**Compliance With Llm Reviewing Policy:**

Affirmed.

**Final Justification:**

The author has answered most of my questions, so I will maintain a high rating.

**Key Questions For Authors:**

### Key Questions For Authors

* **Regarding the "pure neutralization" of counterfactual images and confounder control**:
How do you quantitatively verify that the differences between I_emo and I_neu stem primarily from the "emotion variable" rather than other visual semantic changes (e.g., mouth shapes/frowns are emotional cues themselves, but they might also alter identity details, perceived age, or scene details)? Could you provide: statistics on general visual/identity similarity (e.g., face embeddings, CLIP image embedding similarity) before and after counterfactual generation; or evidence using "non-emotional tasks" (object/attribute/identity consistency) to prove that the pairings remain unchanged?
* **Regarding the universality of the mechanism conclusions (more complex/more emotion categories)**:
Currently, the mechanism analysis mainly covers basic emotions, and the paper acknowledges potential shortcomings for complex emotions. Could you supplement whether the functional decoupling of "middle-layer emotion-specific aggregation — deep-layer emotion-general execution" still holds on more fine-grained or mixed emotion samples (such as subsets of OV-MERD+)? If it holds, it indicates the mechanism is more universal; if not, it is recommended to explicitly state the applicable boundaries of this mechanism and future directions.
* **Regarding the safety and side-effect profile of VEENA**:
You showed that on general capability benchmarks, there are "not always gains and some trade-offs." Could you systematically analyze: Does VEENA lead to "over-emotionalization/emotional bias" (e.g., a higher tendency to describe even neutral images as sad or angry), and is this bias amplified under different prompting styles or decoding strategies? Proving that side effects are controllable and have clear on/off switches or intensity adjustment strategies would enhance its value for practical deployment; if side effects are prominent, more cautious conclusions and positioning are needed.

**Limitations:**

Yes

**Strengths And Weaknesses:**

### Strengths

* **Acutely targeted and practically significant problem definition**: It addresses "emotional hallucinations" as a reliability issue related to safety and human-computer interaction, rather than merely stacking emotion classification performance.
* **Complete methodological chain**: The progression from mechanism discovery to actionable intervention, and finally to benchmark validation, forms a relatively complete logical closed loop. It not only localizes the circuits but also uses VEENA for "surgical" inference-time interventions and validates the benefits on benchmarks.
* **More suitable metrics for "open-ended descriptive emotional reasoning"**: The authors recognize that token-level logit differences are unsuited for long-text emotional expressions, and instead design latent restoration/hidden-space metrics to measure the causal contributions of modules, which is a reasonable and well-justified direction.
* **Mechanistic conclusions possess explanatory power**: The functional decoupling of "emotion-specific aggregation" in the middle layers and "emotion-general execution" in the deep layers provides a natural justification for why two types of interventions ("routing enhancement + neuron amplification") can be applied to different stages.
* **Relatively comprehensive experimental coverage**: The paper reports overall gains on MER-UniBench and additionally checks the preservation of general capabilities on the POPE, CHAIR, and MME benchmarks.

### Weaknesses

* **Counterfactual image construction may introduce uncontrollable confounders**: The paper relies on a generative model to "neutralize" emotional images to construct (I_emo, I_neu) pairs. However, even if the prompt emphasizes pixel-level precision, it might still introduce subtle semantic or texture changes in practice, thereby affecting the credibility of the steering vectors and circuit localization.
* **Limited extrapolation of the discovered mechanisms**: The authors primarily use 4 basic emotions for steering and mechanism analysis, which may be insufficient for mixed, subtle, or high-context emotions. Therefore, it remains unclear whether the "emotional pathways" equally apply to a more complex emotion spectrum.

---

> ### Author Rebuttal · Authors · 2026-03-30
>
> We sincerely thank the reviewer for highly positive evaluation and constructive feedback. We commit to incorporating the valuable discussions below into the revised manuscript.
>
> > **W1&Q1：Verification of $(I_{emo}, I_{neu})$**
>
> Isolating the "emotion variable" is paramount for rigorous mechanistic analysis. To verify our generated counterfactual pairs $(I_{emo}, I_{neu})$ introduce no semantic or identity shifts, we conducted comprehensive validations:
>
> 1. **Global Semantic Consistency**
>
> Cosine similarities of global image embeddings (extracted via target LVLM Vision Encoders) show our $(I_{emo}, I_{neu})$ pairs maintain near-perfect visual consistency, whereas random pairs exhibit no similarity.
>
> |Evaluation Set|Qwen3-VL-4B|LLAVA-OV-1.5-4B|Qwen3-VL-8B|
> |:-|:-|:-|:-|
> |$I_{emo}$ vs $I_{neu}$|0.9742|0.9747|0.9742|
> |Random Pairs|-0.1671|-0.7429|-0.1671|
>
> 2. **Facial Identity Preservation**
>
> Using the deepface framework [1], we evaluate the collected human image pairs across four expert models. Identity similarities of our pairs heavily out-margin random baselines, proving core facial features are retained despite neutralized expressions.
>
> |Evaluation Set|VGG-Face|ArcFace|GhostFaceNet|Facenet512|
> |:-|:-|:-|:-|:-|
> |$I_{emo}$ vs $I_{neu}$|0.4540|0.5317|0.3373|0.4943|
> |Random Pairs|0.0661|0.0351|0.0668|0.1951|
>
> 3. **Consistency in Non-Emotional Tasks**
>
> Prompting LVLMs with "Consistency" questions regarding the main Object, background Attribute, and physical Identity (ignoring expressions) confirms the $I_{neu})$ remain identical across non-emotional dimensions with high accuracy.
>
> |Consistency Type|Qwen3-VL-4B|LLAVA-OV-1.5-4B|Qwen3-VL-8B|
> |:-|:-|:-|:-|
> |Object|98.87|100.0|96.59|
> |Attribute (Background)|96.59|98.87|98.87|
> |Identity (Traits/Age)|97.72|100.0|100.0|
>
> 4. **Vector Projection**
>
> Projecting the "Happy" vector (Qwen3-VL-4B) onto the vocabulary space reveals that from layer 19, top-activated tokens transition strictly into pure, abstract emotional concepts, not physical descriptions:
>
> * Layer 18: ['意大', '发扬', 'Alic', '这家',  '因为我'...]
> * Layer 19: ['delight', 'ecstatic', 'joyful', '快乐', 'joy'...]
> * Layer 35: ['joyful', 'joy', '欢呼', '欢乐', 'delighted'...]
>
> > **W2&Q2: Extrapolation to Fine-Grained and Mixed Emotions**
>
> Humans express over 34,000 distinct emotions [2], making exhaustive granular labeling computationally prohibitive [3]. Thus, we follow AffectGPT [4] to evaluate open-ended expressions by hierarchically mapping fine-grained generated emotions (e.g., "Hopeful", "Loving") to core basic anchors (e.g., "happy").
>
> To explicitly verify whether our mechanism extrapolates to these complex emotions, we conducted a phase-level activation patching experiment (similar to **Fig.5(c)**) for the fine-grained emotion "hopeful". Instead of measuring the basic emotion hit rate, we strictly tracked the specific **generation frequency** of the word "Hopeful".
>
> As shown below, the causal trajectory perfectly replicates our "Adapt-Aggregate-Execute" relay, confirming that the functional decoupling universally holds for fine-grained emotional subsets.
>
> |Token|Early|Early-Mid|Mid-Late|
> |:-|:-|:-|:-|
> |Visual|**0.036**|0.022|0.021|
> |Query|0.025|**0.027**|0.032|
> |Last|0.022|0.024|**0.036**|
>
> However, the mixed emotions (e.g., bittersweetness) likely involve more entangled reasoning pathways. We explicitly acknowledge this boundary in our **Limitations** as a vital direction for future work.
>
> > **Q3: Safety and Side-effect Profile**
>
> To systematically verify if VEENA induces "over-emotionalization," we evaluated Qwen3-VL-4B on neutral images ($I_{neu}$) across two tasks, using greedy and sampling decoding (matching **Figure 10** hyperparameters):
>
> 1. Emotion Recognition on Neutral Images
> * Task: Can the model correctly identify neutral states without bias?
> * Prompt: "Is this emotion happy, sad, angry, fear, neutral or else?"
> * Metric: Accuracy (%)
>
> 2. Emotion-Agnostic Image Description
> * Task: Does the model simply generate more emotion words?
> * Prompt: "Please describe this image in detail, focusing on the main subjects and their actions."
> * Metric: LLM-as-Judge Emotion Score (1: Completely objective $\to$ 5: Highly emotional).
>
> Quantitative Results:
>
> |Task (Metric)|Greedy Decoding|Baseline|Baseline+VEENA|
> |:-|:-|:-|:-|
> |Recognition (Acc ↑)|√|98.86|96.59|
> |Recognition (Acc ↑)|×|90.90|88.63|
> |Description (Score ↓)|√|1.43|1.59|
> |Description (Score ↓)|×|1.61|1.69|
>
> Results show VEENA does not trigger over-emotionalization. Accuracy drops on neutral images are marginal, and Emotion Scores during general descriptions remain strictly objective. While effective under fixed intensity, we will discuss dynamic intensity adjustment as future work in the revised Limitations.
>
> > **References.**
>
> [1] Deepface: Closing the gap to human-level ... CVPR 2014.
>
> [2] The nature of emotions: Human emotions ... American scientist 2001.
>
> [3] OV-MER: Towards Open-Vocabulary ... ICML 2025.
>
> [4] Affectgpt: A new dataset, model ... ICML 2025 oral.

---

> > ### Author Rebuttal · Reviewer_UcKy · 2026-04-02
> >
> > The author has answered most of my questions, so I will maintain a high rating.

---

> > > ### Author Response · Authors · 2026-04-02
> > >
> > > Thank you very much for your positive feedback and for maintaining your **high rating (5/6)** of our work.
> > >
> > > Your insightful questions were instrumental in **enhancing the clarity and overall quality of our manuscript**. We are fully committed to incorporating all the clarifications and additional details discussed during this rebuttal phase into the final version of the paper. We are glad to address any further questions you may have.
> > >
> > > **Thank you again for your time and constructive review**.

---

### Official Review · Reviewer_5Qbu · 2026-03-10

**Soundness:** 3
**Presentation:** 3
**Significance:** 3
**Originality:** 3
**Overall Recommendation:** 4
**Confidence:** 2

**Summary:**

This paper presents a representational analysis of sentiment understanding in MLLM. Specifically, it uses the nano banana model to generate similar counterfactual images from images and analyzes the changes in internal representations. It also uses the steering technique to locate sentiment-related representations.

**Compliance With Llm Reviewing Policy:**

Affirmed.

**Final Justification:**

After considering the rebuttal, I acknowledge the authors’ point that their method is not a direct application of existing interpretability techniques. Nevertheless, the overall improvement remains broadly similar to what has been observed in prior work. Although I do not consider this a decisive weakness, especially given the downstream analysis setting, I do not find the rebuttal sufficient to change my overall assessment. Therefore, I will maintain my current score.

**Key Questions For Authors:**

See the weaknesses.

**Limitations:**

The paper includes a limitation section in the appendix.

**Strengths And Weaknesses:**

### Strengths

- It's very interesting to construct counterfactual image inputs using generative models.
- Analyze neurons related to emotion from their internal representations.

### Weaknesses

- The novelty lies mostly in applying them to emotional reasoning, not developing new interpretability methodology. Compared with recent circuit papers, the algorithmic contribution is incremental.
- The evaluation metric uses LLM-as-Judge to extract emotion keywords, then maps them to emotion wheels. If extend human evaluation would strengthen reliability.
- The paper claims to mitigate emotional hallucinations, but the definition is somewhat vague, because emotion hallucination is more subjective.

---

> ### Author Rebuttal · Authors · 2026-03-30
>
> We thank the reviewer for the positive evaluation and valuable suggestions. We will update the relevant discussions in the manuscript accordingly.
>
> > **W1: Not developing new interpretability methodology.**
>
> We respectfully clarify that our contribution is merely an application of existing methods; instead, we propose a novel, steering-vector-based interpretability framework. To clarify these core methodological differences against existing methods, we summarize a table below:
>
> |Difference|Existing mechanistic interpretability framework [2,3]|Our Steering-vector-based Framework|
> |-|-|-|
> |1. Causal Effect Metric|Discrete Next-Token-Predict Logits|Continuous Latent-Space similarity|
> |2. Tracing Anchor|Strictly restricted to the final output layer|Flexible backtracing from intermediate layer|
> |3. Information Routing|Isolation between heads and Neurons|Holistic *Head-Token-Neuron* interactive relay|
> |4. Scope of application|Single-word factual entities (e.g., object recognition, 'True' or 'false' questions)|Diffusive, long-form narratives (e.g., emotion understanding, LLM personality, and role-playing)|
>
> Specifically, the diffuse nature [1] of emotional reasoning fundamentally breaks traditional interpretability methods, necessitating our novel methodological designs. Our algorithmic contributions diverge from recent circuit papers [2,3] in four fundamental ways:
>
> 1. **Causal Effect Metric**: Prior methods rely on discrete token probabilities. However, emotion is inherently diffusive and complex, **which cannot be captured by a single target word**. Our framework overcomes this by introducing a vector-based Latent Restoration Metric, shifting the causal evaluation from discrete output logits to continuous hidden spaces.
>
> 2. **Tracing Anchor**: Standard logit-based patching relies on final output probabilities, rigidly restricting causal tracing to start only from the final output layer. In contrast, our vector-based paradigm allows causal backtracing from any intermediate layer. This architectural freedom is exactly what enabled us to uncover the stage-wise "Adapt-Aggregate-Execute" behavioral transitions.
>
> 3. **Information Routing**: Rather than isolating individual attention heads or neurons, our framework systematically maps the complete relay—linking attention flow (heads), information routing (tokens), and semantic knowledge (neurons)—to explain complex inter-module interactions.
>
> 4. **Scope of application**: Furthermore, our proposed framework is not restricted to emotion understanding. It serves as a generalized, scalable tool for evaluating descriptive generation tasks where traditional NTP metrics fail. This includes analyzing model persona alignment and role-playing.
>
>
> > **W2: Extending to human evaluation.**
>
> First, we clarify that the open-ended evaluation metric (*hit rate*) **was not proposed** by us, but strictly follows the established protocol of AffectGPT (ICML 2025 Oral) [4]. In our mechanistic analysis, this metric serves solely as an automated filtering tool to select high-quality contrastive pairs for extracting steering vectors.
>
> **Added Human Evaluation:**
>
> To further validate the reliability of this metric, we conducted a supplementary human evaluation. We randomly sampled 100 "image + model response" pairs from our dataset: 50 accepted by our metric (used for vector extraction) and 50 filtered out.
> We recruited 3 psychology students to blindly judge whether the generated responses accurately reflected the target emotions. The human judgments achieved 98% and 96% agreement with our automated filtering results. This high consistency strongly corroborates that our metric-based screening is highly reliable for isolating valid emotional pathways.
>
> > **W3: Definition of Emotion Hallucination.**
>
> Recent **EmotionHallucer (ICLR 2026)** [5] systematically categorizes emotional hallucinations into "Emotion Knowledge Hallucination" (errors regarding emotion theories and definitions) and "**Multimodality Perception Hallucination**" (errors in emotion recognition and reasoning). The hallucinations targeted in our work strictly fall into this **latter category**.
>
> While emotion inherently involves subjectivity, "hallucination" in the context of LVLM has a clear definition: generating factually inconsistent outputs that contradict the provided visual inputs. Rather than inventing a new concept, we treat "emotional hallucination" as a specific subset of this broader challenge, much like objective "object hallucination".
>
> > **References.**
>
> [1] Relations among emotion, appraisal, and emotional action readiness. JPSP 1989.
>
> [2] Same Task, Different Circuits: Disentangling Modality-Specific Mechanisms in VLMs. NeurIPS 2025.
>
> [3] Vision-Language Models Create Cross-Modal Task Representations. ICML 2025.
>
> [4] Affectgpt: A new dataset, model, and benchmark for emotion understanding... ICML 2025 Oral.
>
> [5] Emotionhallucer: Evaluating emotion hallucinations in multimodal large language models. ICLR 2026.

---

> > ### Author Rebuttal · Reviewer_5Qbu · 2026-04-02
> >
> > Thank you for the author's reply. After reading the rebuttal, I understand the author's claim that they didn't directly use existing interpretable methods, but the overall improvement is still quite similar to existing methods (of course, this isn't a fatal problem in downstream task analysis). After careful consideration, I've decided to maintain the current score.

---

> > > ### Author Response · Authors · 2026-04-02
> > >
> > > **Thank you for your continued engagement and for maintaining your positive evaluation**. We deeply appreciate your time and candor. To address your remaining reservations, we feel it is crucial to further clarify the fundamental methodological advancement our framework introduces to Mechanistic Interpretability (MI), which extends far beyond a simple application to a new domain.
> > >
> > > To clearly articulate this algorithmic distinction, we summarize the specific gap we address and how our method solves it:
> > >
> > > 1. **Existing Problem: The Single-Token Bottleneck**. There is a fundamental gap in existing MI methods [1,2,3,4,5,6]: they are inherently built around single-token prediction objectives. Consequently, they lack a suitable metric for tracing diffuse, long-form generative behaviors, making the causal analysis of complex, multi-token completions a recognized open challenge in the field.
> > >
> > > 2. **Our Method: A Continuous Multi-Token Proxy**. Our proposed continuous Latent Restoration Metric directly tackles this bottleneck. By using the steering-vector-based emotional direction as a continuous *multi-token proxy*, we decouple causal attribution from discrete Next-Token Prediction (NTP) logits. This critical shift—evaluating causal effects in the *hidden latent space* rather than relying on discrete output probabilities—successfully bypasses the single-token limitation.
> > >
> > > 3. **The Significance: Unlocking Generative Behaviors**. This methodological leap makes it possible to trace subjective concepts like "emotion" across open-ended, long-form narratives—a capability currently beyond the reach of standard MI methods. Furthermore, it makes our activation patching framework applicable not only to emotion understanding but also to other diffuse generative behaviors (e.g., persona alignment and role-playing).
> > >
> > > We will ensure this specific distinction—solving the multi-token limitation for generative tasks—is prominently highlighted in the revised manuscript to better clarify our methodological contribution.
> > >
> > > **Thank you again for your constructive reviews and for helping us refine the positioning of our work.**
> > > We sincerely hope these explanations fully resolve your remaining concerns, and earn your favorable evaluation.
> > > we would be deeply grateful if you might reconsider your overall evaluation of our work, and glad to provide additional clarifications for any further questions.
> > >
> > > > **References.**
> > >
> > > [1] Same Task, Different Circuits: Disentangling Modality-Specific Mechanisms in VLMs. NeurIPS 2025.
> > >
> > > [2] Vision-Language Models Create Cross-Modal Task Representations. ICML 2025.
> > >
> > > [3] Circuit Tracing: Revealing Computational Graphs in Language Model. Transformer Circuits Thread, 2025
> > >
> > > [4] Towards Best Practices of Activation Patching in Language Models: Metrics and Methods. ICLR 2024.
> > >
> > > [5] Interpretability in the wild: a circuit for indirect object identification in gpt-2 small. ICLR 2023.
> > >
> > > [6] How does GPT-2 compute greater-than?: Interpreting mathematical abilities in a pre-trained language model. NeurIPS 2023.

---

### Official Review · Reviewer_sJsR · 2026-03-10

**Soundness:** 3
**Presentation:** 2
**Significance:** 3
**Originality:** 2
**Overall Recommendation:** 4
**Confidence:** 3

**Summary:**

The authors introduce a mechanistic interpretability technique catered towards the emotional understanding of vision-language models. The pipeline is catered to natural language generations, evaluating narratives instead of simple classification. The focus is on how visual stimuli related to emotion are processed and assimilated into the linguistic space of the model, showing that targeted interventions can enhance the emotional information extracted from the image/videos, reporting stronger emotional signals within the generated narratives.

**Compliance With Llm Reviewing Policy:**

Affirmed.

**Final Justification:**

While I currently find the paper very hard to read, the authors have tried to clarify many concepts in the rebuttal. They also provided/will provide extra baselines which are critical for readers. Given the recommendations from other reviewers as well, I have raised my score.

**Key Questions For Authors:**

The weaknesses of the paper are presented in the form of questions already above, but I think it's useful to summarize some of the main points here again:
- Why are no baselines presented to compare with your method?
- Why is it positioned as a training-free algorithm? Gradients, differences, and labels are already used widely within the algorithm, thus training seems like a reasonable and much simpler alternative
- Why were classification tasks left out of the paper, with a focus on generation, which is evaluated based on keywords?
- What useful information does the mechanistic analysis yield? Why is it useful above and beyond moving the needle in the evaluations?

**Limitations:**

Yes

**Strengths And Weaknesses:**

- Presentation: I want to start with the presentation, since most everything else follows from that. The paper is poorly written, with many central concepts unexplained in the main text, and cryptic or missing even from the appendix, requiring multiple passes to even make some sense of things that are supposed to orient the reader. For example, the abstract was unintelligible. I had to make a first pass over the paper to then understand the abstract. The same goes for the introduction. Moreover, after reading the paper, some of the appendix, and Section 2 multiple times, I am still not clear on what the preliminaries mean. Does the LLM judge extract emotional keywords from the generated text or for the specific expected emotion? If the former, why the fixed cardinality of the set $N_p$? And why is it an "LLM-as-a-judge" if it is not judging? What are "standardized" emotion wheels, how's the mapping and evaluation done on them, and why $N_w$ of them? In the next paragraph, for activation patching, why do the authors first declare that "[t]o identify the critical components [...], we leverage [...] Activation Patching", but then at the end of the section declare activation patching to be "ill-suited"? In 3.1, what are "neutral events" (there is some definition in the Appendix but the authors don't cite that)? Why is it "random"? Why is the text neutral too? My guess is that we don't want textual emotion information, only visual, but it is not explained. Why is the counterfactual generation process not even mentioned in the main text? Why the last token positionS $N$? Is $N$ a set of the last few tokens, or is it supposed to be "position" (singular)? What exactly is $h$? And so on for the rest of the paper.
- Soundness: Given the inherent difficulties in understanding the paper, it is difficult to judge the soundness of the paper. However, I do have one central concern with the soundness of the findings. The authors implicitly compare their algorithm to training-based algorithms, calling it a training-free intervention. However, it is actually neither unsupervised, and gradients or differences (can be seen as gradients without the limit to 0) are indeed being used throughout the algorithm. Why not train then? And how does your approach compare to other interventions, like LoRA, in the test set? Why is only zero-shot being compared to the supervised methods? Multimodal LLMs are known to follow instructions poorly, so a zero-shot baseline might be failing due to poor instruction following. This confounding factor has not been examined: does steering just incentivize the production of more emotional words? Useful finding even if so, yet I perceive it as different from the claims in the paper. Finally, the improved performance is discussed as mitigation of emotional hallucinations, which I think is trivially true, but not informative. Any misclassification can be seen as hallucination.
- Significance: The problem is interesting and important, but the framing does not adequately capture that. Beyond small improvements in performance, why is this mechanistic understanding useful in LLMs? The authors should try to strengthen that part of the paper.
- Originality: extending mechanistic interpretability to a subjective domain comes with its own set of challenges. To support their claims of novelty, the authors should try to establish more if and what is different from contemporary approaches, used in object detection as the authors cite for example.

---

> ### Author Rebuttal · Authors · 2026-03-30
>
> We appreciate the reviewer's feedback. As most concerns stem from **misunderstandings** of standard Mechanistic Interpretability (MI) and Multimodal Emotion Recognition (MER) techniques, we respectfully clarify these basic concepts first:
>
> 1. **Activation Patching and Why it is "Ill-suited" for Neurons**: Activation Patching [1] is a standard causal intervention technique in MI used to localize critical model components. However, it requires brute-force sweeping for single component **one-by-one**. While feasible for attention heads, it is computationally intractable and thus "ill-suited" for localization across **millions of MLP neurons**.
>
> 2. **Attribution Patching (Gradients) vs. Model Training**: To solve the computational bottleneck, Attribution Patching [2] uses gradients of intermediate hidden activations—not model weights—to efficiently estimate causal effects. The model parameters remain strictly frozen throughout the entire process.
>
> 3. **Emotion Wheels**: We followed **AffectGPT (ICML 2025 Oral)**[3] and **OV-MER (ICML 2025)**[4]. They use 5 wheels to map open-world emotions. The fine-grained emotions (e.g., "Hopeful" or "Powerful") can be hierarchically mapped to a core basic emotion (e.g., "Happy") via these wheels. This tool therefore extends closed-set emotion recognition (<10 categories) to open-world (>200).
>
> > **W1: Presentation and Methodological Details**
>
> Due to strict 8-page limit, we may only **cited** some widely used core concepts in the main text, while some details were deferred to the **Appendix**. Below is a point-by-point clarification:
>
> |Issue|Response|
> |-|-|
> |Target of LLM extraction|The LLM extracts keywords from the model's long-form generated text, not the expected ground truth.|
> |Meaning of $N_p$|$N_p$ is not a fixed global constant; it simply denotes the variable number of keywords extracted from a given response.|
> |"LLM-as-Judge"|It extracts complex emotions, as emotions can be expressed even **without** explicit vocabulary (e.g., via actions).|
> |Emotion wheels|See the point 3 above.|
> |"Ill-suited"|See the point 1 above.|
> |Neutral text|the text must be strictly neutral to prevent textual confounders, ensuring we isolate circuits driven purely by visual cues.|
> |"Random" selection|Neutral events are randomly selected to ensure the extracted emotional steering vectors do not overfit to specific textual contexts.|
> |Counterfactuals in Appendix|Due to the space limit, the pipeline for data generation was placed in **Appendix B (cited at Section 4.1)**. |
> |Meaning of $N$ and $h$|As stated in **Section 3.1**, $N$ denotes the last input token position. $h$ denotes the activation.|
>
> > **W2 (including Q2): Soundness**
>
> |Issue|Response|
> |-|-|
> |"Training-free" claim|We use emotion labels and activation gradients only during the analysis phase to locate emotion circuits. During downstream application, we intervene on specific components during inference with frozen parameters; thus, it is strictly "training-free".|
> |Comparing with LoRA|In fact, **LoRA is not an intervention method** [5]; it's a supervised training method requiring high-quality target datasets to update weights [6]. This comparison is **unfair and completely out of the scope** of our research.|
> |Inducing more emotional words?|No. See our responses to **Q3 of Reviewer UcKy**|
> |Emotional Hallucination|See our response to **W3 of Reviewer 5Qbu**|
>
> > **W3 (including Q4): Significance**
>
> MI is a very popular research topic and moves beyond "black-box" training to uncover how LLM/LVLM works.
>
> As **S1 of Reviewer UcKy**, addressing emotional hallucination is a critical "reliability issue related to safety and human-computer interaction." Besides, our analysis results lays the inspiration for pin-point fine-tuning (**Section 4.3**).
>
> > **W4: Originality.**
>
> We have provided sufficient evidence to explain that we have proposed a novel MI analysis framework (**W1 of Reviewer 5Qbu**).
>
> In addition, as **S4 of Reviewer UcKy**, we verified on the general dataset (object recognition) that our VEENA did not lose the general ability of the model.
>
> > **Q1: Lack comparisons.**
>
> Our primary contribution is proposing a novel MI framework to interpret emotion circuits. We have further added comparisons with SOTA intervention methods (**Q4 of Reviewer dofs**.)
>
> > **Q3: Why focus on generation.**
>
> As explicitly stated in our **Introduction** and **S2 of Reviewer UcKy**, we omitted classification tasks because single-word labels are fundamentally inadequate for capturing the complex, diffusive nature of emotions [7].
>
> > **References**
>
> [1] Towards Best Practices of Activation Patching ... ICLR 2024.
>
> [2] Attribution Patching Outperforms Automated ... BlackboxNLP 2024.
>
> [3] Affectgpt: A new dataset, model, and ... ICML 2025 oral.
>
> [4] OV-MER: Towards Open-Vocabulary ... ICML 2025.
>
> [5] Inference-time intervention ... NIPS 2023.
>
> [6] Lora: Low-rank adaptation of large ... ICLR 2022.
>
> [7] Relations among emotion, appraisal ... JPSP 1989.

---

> > ### Author Rebuttal · Reviewer_sJsR · 2026-04-02
> >
> > I appreciate the rebuttal in many respects, especially for providing baselines and some explanations. I do believe some of the concerns I and other reviewers raised have not been argued as much as they should, with citations used instead (similarly to the paper, as I point out in my original review, even the rebuttal is not as self-contained as possible).
> >
> > The main concern with the presentation of the paper still remains, and the rebuttal only partly addressed that. For example, I am well-aware where and how $h$ was defined. As I mentioned, I read many sections of the paper multiple times. The review meant to signify that the description in the text, as currently presented, is inadequate. That makes references to the main text within the rebuttal also inadequate. Apologies for any misunderstanding.
> >
> > In conclusion, the inclusion of baselines, along with hopefully some clarifications in the writing, and the opinions of the rest of the reviewers, have improved my opinion of the paper. I do unfortunately find that the main weakness of the paper, its presentation, has not been adequately addressed. Perhaps the authors can provide a write up that clarifies some or most of the points raised in the original review that have not been addressed.

---

> > > ### Author Response · Authors · 2026-04-02
> > >
> > > **We sincerely appreciate your careful rereading and detailed follow-up**. Your feedback rightly highlights the need for clearer, self-contained explanations of several key concepts, rather than simple pointers to definitions. We therefore focus on directly addressing and clarifying these remaining conceptual points below.
> > >
> > > * **Evaluation.** Our task is not closed-set classification, but emotion-grounded narrative generation. Accordingly, the auxiliary LLM is used only to read the generated response and extract a small set of emotion keywords that summarize its emotional tone; it is not used to decide the target label itself. We also agree that the phrase ''LLM-as-a-Judge'' is potentially confusing in this context; in the revision, we will instead describe it more directly as an auxiliary LLM used to extract emotion keywords from the generated response. These free-form keywords are then mapped to standardized emotion wheels—i.e., a fixed hierarchical taxonomy used to normalize open-vocabulary affect terms. Here, $N_w$ denotes the number of wheels; following AffectGPT, we use five wheels. For each wheel, we check whether the mapped target emotion is contained in the mapped keyword set, and the final hit score is the average of these five wheel-specific hits. Likewise, $N_p$ is not a fixed global constant; it simply denotes the number of emotion keywords extracted for a given response. Our goal is to analyze how visual emotion is translated into narrative expression, which is precisely the part that single-label classification suppresses.
> > >
> > > * **Control setup.** In Section 3.1, the neutral text and randomly sampled neutral events are used to remove textual emotional priors and avoid overfitting the extracted direction to a single fixed context; the goal is to isolate **visually driven** emotional variation as much as possible. Likewise, the counterfactual image generation step is not meant to be a hidden assumption, but to construct pairs that differ primarily in emotional expression rather than scene semantics. While the detailed pipeline was deferred to Appendix B, we agree that this control step is central enough that the main text should have included a brief summary, and we will add one in the revision. Also, ''last token positions'' was imprecise wording on our side: we mean the *last input token position* (singular). Specifically, $h_{i,l,N} ∈ R^d$ denotes the residual-stream activation vector for sample $i$ at layer $l$ and the position $N$.
> > >
> > > * **Activation patching and ''training-free.''** On activation patching, our intended point was about *scalability rather than validity*. We use activation patching as a causal tool where it is tractable, especially for layers and heads. What is ''ill-suited'' is exhaustive neuron-level search with brute-force patching, because that becomes computationally impractical at the scale of MLP neurons. Similarly, our use of ''training-free'' was narrower than it should have been: we do not mean that the full discovery pipeline is unsupervised or gradient-free. We mean that, once the relevant circuit is identified offline, VEENA is applied at inference time *without updating model parameters*. Therefore, we further added comparisons with other inference-time interventions rather than parameter-updating methods such as LoRA.
> > >
> > > * **What VEENA claims.** We agree that a stronger emotional style alone would be a different claim from better *visual-emotion grounding*. Our intent is not to simply induce more emotional wording, but to improve alignment between visual affective cues and the generated narrative. For this reason, the claim should be stated more carefully: VEENA aims to reduce *emotionally misalignment*, rather than treating every error as ''hallucination'' in an undifferentiated sense.
> > >
> > > * **Why the mechanistic analysis matters.** The mechanistic analysis is meant to do more than justify a score improvement. Its main value is that it reveals a functional separation: middle-layer components are more involved in aggregating and routing visual emotional cues, while deeper components are more involved in converting those cues into explicit emotional semantics for generation. VEENA was designed from that separation: one part strengthens information routing, and the other reinforces semantic activation. More broadly, our novelty is not simply to transplant object-centric circuit analysis to emotion, but to adapt causal probing to a subjective, open-ended generation setting where token-level logit-based evaluation is insufficient and semantically controlled visual counterfactuals are required.
> > >
> > > **We appreciate your emphasis on presentation**. In the revision, we will integrate these parts directly into the main text instead of relying on citations. Furthermore, we will revise the abstract and introduction to explicitly state our task, evaluation protocol, and contributions.
> > >
> > > **We are more than willing to address any further questions**.

---

### Official Review · Reviewer_dofs · 2026-03-11

**Soundness:** 3
**Presentation:** 2
**Significance:** 3
**Originality:** 3
**Overall Recommendation:** 4
**Confidence:** 2

**Summary:**

This paper studies the internal mechanisms of emotion understanding in large vision-language models by introducing a steering-vector-based causal attribution framework for descriptive emotional reasoning. It identifies a hierarchical “Adapt-Aggregate-Execute” mechanism and reveals a functional decoupling in which emotion-specific heads aggregate visual emotional cues in middle layers, while emotion-general pathways support downstream narrative generation. Based on these findings, the paper further proposes VEENA, a training-free inference-time intervention that modulates emotional information routing and semantic activation.

**Compliance With Llm Reviewing Policy:**

Affirmed.

**Final Justification:**

I appreciate the authors’ detailed response. Most of my concerns have been addressed, and I therefore maintain my initial positive assessment of the paper. I encourage the authors to further revise the manuscript by incorporating the rebuttal content.

**Key Questions For Authors:**

Please refer to Weaknesses.

**Limitations:**

yes

**Strengths And Weaknesses:**

### Strengths

1. The paper is well motivated, clearly organized, and easy to follow.
2. The identified hierarchical “Adapt-Aggregate-Execute” mechanism offers an interesting perspective for understanding how large vision-language models process emotions.
3. Extensive experiments and analyses support the paper’s findings on emotional circuits and internal mechanisms, while also validating the effectiveness of the proposed VEENA.

### Weaknesses

1. The definition of Multi-modal Emotion Recognition in Section 2 (Preliminaries) appears somewhat imprecise. In general, MER typically refers to settings where the input involves multiple modalities such as vision, audio, and text.
2. The specific meaning and design rationale of Equation (10) should be explained more clearly, as the current presentation is too brief.
3. The paper investigates general-purpose MLLMs as baselines, such as Qwen3-VL-4B-Instruct, Qwen3-VL-8B-Instruct, and LLaVA-OneVision-1.5-4B-Instruct. It would be valuable to discuss whether the conclusions would remain the same for emotion-specialized MLLMs fine-tuned on emotion-related instruction-following datasets, such as AffectGPT, and how VEENA performs on such models.
4. The paper lacks comparisons between VEENA and other representative inference-time steering or intervention methods, which limits an intuitive demonstration of the advantages of the proposed design.
5. The paper claims that VEENA improves performance without additional latency, but lacks computational analysis or runtime comparisons to substantiate this claim.
6. Extracting the emotional directions only across four basic emotions may be limited when extending to broader fine-grained emotion analysis.
7. The text in the right column around line 193 appears to be incomplete.

---

> ### Author Rebuttal · Authors · 2026-03-30
>
> We deeply appreciate the reviewer for positive remarks and constructive guidance. We will certainly reflect these valuable points in the revised manuscript.
>
> > **W1: Definition of MER.**
>
> MER is a process that integrates multiple data modalities such as speech, visual, and text to identify human emotions [1]. We defined our research scope (visual and text) in the **Preliminaries** and discussed broader modalities (audio) in the **Limitations**. We will further clarify this definition.
>
> > **W2: Clarification of Equation (10).**
>
> To clearly connect the mathematical formulation with our mechanistic findings, we will expand the explanation below Equation (10) to clearly detail its specific meaning and rationale as follows:
>
> > Crucially, $\mathcal{P}\_{up}$ triggers during the prefill phase (t=0) at upstream layers, amplifying V $\to$ Q attention to facilitate the Aggregation of visual emotional cues into the textual query. Conversely, $\mathcal{P}\_{down}$ activates during the decoding phase (t>0) at downstream layers, strengthening V $\to$ L attention to ensure the narrative Execution remains firmly grounded in fine-grained visual details.
>
> > **W3: Extending to emotion-specialized MLLMs.**
>
> While a comprehensive extension to emotion-specialized MLLMs (audio/video) is left for future work, we provide preliminary results on Emotion-Qwen [2] to demonstrate its feasibility. This expert model maintains strong general vision-language capabilities and, like AffectGPT, utilizes the Qwen2.5-Instruct-7B backbone. Note that we only focus on image and text modalities.
>
> Firstly, similar to **Figure 5** in the main text, we traced the information flow and divided it into Early (0-10), Early-Mid (11-21), and Mid-Late (22-28) based on layers, verifing via phase-level activation patching as below (reports the change ratio of hit rate). These results show that the three-stage mechanism still exists.
>
> |Token|Early|Early-Mid|Mid-Late|
> |:-|:-|:-|:-|
> |Visual|**0.3824**|0.2129|0.2796|
> |Query|0.1731|**0.4611**|0.3879|
> |Last|0.0685|0.2879|**0.4231**|
>
> Then, we apply function verification of heads similar to **Figure 7** in the main text and observed consistent phenomena below: disrupting (Zero) or recovering (Restore) the model's behavior (reports the hit rate) is only significant when intervening on the specific emotion-related heads.
>
> |methods|baseline|Top-10 / Random-10|Top-20 / Random-20|
> |:-|:-|:-|:-|
> |zero|0.2746|0.1240 / 0.2266|0.1186 / 0.2320|
> |restore|0.1080|0.1586 / 0.0573|0.1880 / 0.0333|
>
> Finally, we evaluated VEENA on diverse MER tasks using this model. As shown below, VEENA remains highly effective.
>
> |Method|MER2023|CMUMOSI|OV-MERD+|
> |:-|:-|:-|:-|
> |Emotion-Qwen|59.70|51.80|40.79|
> |Emotion-Qwen + VEENA|63.85|54.07|43.52|
>
> > **W4: Lack comparisons with other inference-time steering or intervention methods.**
>
> To further demonstrate the advantages of our design, we supplemented our experiments on Qwen3-VL-4B by comparing VEENA against recent SOTA inference-time methods: VISTA (steering, ICML 2025) [3] and PAI (intervention, ECCV 2024) [4]. We ensured all evaluations strictly followed identical experimental settings, including prompts and decoding settings.
>
> |Methods|MER2023|CMUMOSI|OV-MERD+|
> |:-|:-|:-|:-|
> |Baseline|53.92|51.65|40.42|
> |VISTA(ICML 2025)|52.94|53.84|40.82|
> |PAI(ECCV 2024)|59.61|54.37|46.32|
> |VEENA(ours)|**66.03**|**54.97**|**49.67**|
>
> As shown in the table above, VEENA consistently outperforms these baselines across diverse MER tasks. We attribute this superiority to the fact that VEENA is explicitly grounded in our discovered emotion mechanism, providing a precise and interpretable intervention rather than a generalized, black-box adjustment.
>
> > **W5: Lack Latency Analysis.**
>
> We understand the concern regarding latency. We clarify that latency (**ms/token**) ablations are already provided in **Table 1** of the main text. To further demonstrate the high efficiency of VEENA, we provide a more comprehensive analysis on a RTX4090 GPU across different models below. VEENA introduces a negligible latency (e.g., 23.02 vs. 23.57, nearly ×1.02).
>
> |Precise Setting|Qwen3-VL-4B|LLAVA-OV-1.5-4B|Qwen3-VL-8B|
> |:-|:-|:-|:-|
> |Baseline|14.52 |14.62|23.02|
> |+ VEE (Fig.3(1))|14.78|15.05|23.33|
> |+ ENA (Fig.3(2))|14.75|14.69|23.22|
> |+ VEENA (VEE&ENA)|15.06|15.27|23.57|
>
> > **W6: Extending to broader fine-grained emotion analysis.**
>
> Due to strict rebuttal space limitations, please refer to our detailed response to the **W2 of Reviewer 5Qbu**. We apologize for any inconvenience this may cause.
>
> > **W7: Incomplete text in line 193.**
>
> We apologize for this clerical typo. We will correct this error in the revised version.
>
> > **References.**
>
> [1] A Comprehensive Review of Multimodal Emotion Recognition ... Biomimetics 2025.
>
> [2] Emotion-Qwen: A Unified Framework for Emotion ... Arxiv 2505.
>
> [3] The hidden life of tokens: Reducing hallucination ... ICML 2025.
>
> [4] Paying More Attention to Image: A Training-Free Method ... ECCV 2024.

---

> > ### Author Rebuttal · Reviewer_dofs · 2026-04-04
> >
> > Thank you for the detailed response. Most of my concerns have been addressed. I encourage the authors to further revise the paper based on the rebuttal content.

---

> > > ### Author Response · Authors · 2026-04-05
> > >
> > > **Thank you very much for your further response and for maintaining a positive evaluation of our work**. We are truly delighted to learn that our rebuttal has fully resolved your concerns. Your constructive comments and insightful suggestions have been instrumental in improving the overall quality of our manuscript. As you encouraged, we will certainly incorporate the detailed explanations and additional results from our rebuttal into the revised paper to make it more comprehensive.
> > >
> > > Since we have adequately addressed your remaining concerns, we would be deeply grateful if you might reconsider your evaluation of our work.
> > >
> > > **Thank you once again for your valuable time, effort, and guidance throughout this review process**.

---

### Decision · Program_Chairs · 2026-04-30

**Decision:**

Accept (regular)

**Comment:**

The paper presents a novel and well-motivated framework for analyzing and improving emotional reasoning in LVLMs, combining mechanistic insights with a practical inference-time intervention (VEENA). Reviewers pointed out the clear problem formulation, the coherent pipeline from analysis to intervention, and the strong empirical results demonstrating both improved performance and interpretability. The rebuttal effectively addressed most concerns, including additional comparisons, clarification of methodology, and validation of key assumptions, leading to sustained or improved positive evaluations. Overall, the paper offers a meaningful contribution to interpretability and reliability in multimodal models, and is suitable for acceptance.